



# Comparison of three aerosol representations of NHM-Chem (v1.0) for the simulations of air quality and climate-relevant variables

Mizuo Kajino[1,2], Makoto Deushi[1], Tsuyoshi Thomas Sekiyama[1], Naga Oshima[1], Keiya Yumimoto[3,1], Taichu Yasumichi Tanaka[1], Joseph Ching[1], Akihiro Hashimoto[1], Tetsuya Yamamoto[1], Masaaki Ikegami[4], Akane Kamada[4,1], Makoto Miyashita[4], Yayoi Inomata[5,1], Shin-ichiro Shima[6], Pradeep Khatri[7], Atsushi Shimizu[8], Hitoshi Irie[9], Kouji Adachi[1], Yuji Zaizen[1], Yasuhito Igarashi[10,11], Hiromasa Ueda[12], Takashi Maki[1], and Masao Mikami[13,1]

[1]Meteorological Research Institute, Japan Meteorological Agency, Tsukuba, 305-0052, Japan
[2]Faculty of Life and Environmental Sciences, University of Tsukuba, Tsukuba, 305-8572, Japan
[3]Research Institute for Applied Mechanics, Kyushu University, Kasuga, 816-8580, Japan
[4]Japan Meteorological Agency, Chiyoda, 100-8122, Japan
[5]Institute of Nature and Environmental Technology, Kanazawa University, Kanazawa, 920-1192, Japan
[6]Graduate School of Simulation Studies, University of Hyogo, Kobe, 650-0047, Japan
[7]Center for Atmospheric and Oceanic Studies, Graduate School of Science, Tohoku University, Sendai, 980-8578, Japan
[8]National Institute for Environmental Studies, Tsukuba, 305-8506, Japan
[9]Center for Environmental Remote Sensing, Chiba University, Chiba, 263-8522, Japan
[10]Institute for Integrated Radiation and Nuclear Science, Kyoto University, Kumatori, 590-0494, Japan
[11]College of Science, Ibaraki University, Mito, 310-8512, Japan
[12]Disaster Prevention Research Institute, Kyoto University, Uji, 611-0011, Japan
[13]Japan Meteorological Business Support Center, Chiyoda, 101-0054, Japan

*Correspondence to*: Mizuo Kajino (kajino@mri-jma.go.jp)

**Abstract.** This study provides comparisons of aerosol representation methods incorporated into a regional-scale nonhydrostatic meteorology-chemistry model (NHM-Chem). Three options for aerosol representations are currently available: the 5-category nonequilibrium (Aitken, soot-free accumulation, soot-containing accumulation, dust, and sea salt), 3-category nonequilibrium (Aitken, accumulation, and coarse), and bulk equilibrium (submicron, dust, and sea salt) methods. The 3-category method is widely used in three-dimensional air quality models. The 5-category method, the standard method of NHM-Chem, is an extensional development of the 3-category method and provides improved predictions of regional climate by implementing separate treatments of light absorber and ice nuclei, namely, soot and dust, from the accumulation and coarse mode categories. The bulk equilibrium method was also developed for operational air quality forecasting with simple aerosol dynamics representations. The total CPU times of the 5-category and 3-category methods were 91% and 44% greater than that of the bulk method, respectively. The bulk equilibrium method was shown to be eligible for operational forecast purposes, namely, the surface mass concentrations of air pollutants such as $O_3$, mineral dust, and $PM_{2.5}$. The simulated surface concentrations and depositions of bulk chemical species of the 3-category method were not significantly different from those of the 5-category method. However, the internal mixture assumption of soot/soot-free and dust/sea salt particles in the 3-category method resulted in significant differences in the size distribution and hygroscopicity of the



particles. The unrealistic dust/sea salt complete mixture of the 3-category method induced significant errors in the prediction of the mineral dust-containing CCN, which alters heterogeneous ice nucleation in cold rain processes. The overestimation of soot hygroscopicity by the 3-category method induced errors in the BC-containing CCN, BC deposition, and light-absorbing AOT (AAOT). Nevertheless, the difference in AAOT was less pronounced with the 3-category method because the overestimation of the absorption enhancement was compensated by the overestimation of hygroscopic growth and the consequent loss due to in-cloud scavenging. In terms of total properties, such as aerosol optical thickness (AOT) and cloud condensation nuclei (CCN), the results of the 3-category method were acceptable. To evaluate the significance of separate soot and dust treatments in the 5-category method in terms of aerosol-cloud-radiation interaction processes, online simulation with a chemistry-to-meteorology feedback process is required.

# 1 Introduction

Atmospheric aerosols scatter and absorb solar (shortwave) and thermal (longwave) radiation and thus contribute to warming and cooling on regional to global scales. Aerosols also affect regional to global climate in various ways through aerosol-cloud-radiation interactions (Boucher et al., 2013; Oshima et al., 2020). Increases in aerosols can both decrease precipitation from shallow clouds and increase precipitation from invigorated convective clouds as a result of their cloud condensation nuclei (CCN) activities. At the same time, an increase in aerosols enhances atmospheric stability and suppresses convective precipitation as a result of their radiative activities cooling the ground surface and heating the atmosphere (Rosenfeld et al., 2008). Aerosols together with the relevant gases can have negative impacts on ecosystems and health due to the production of photochemical oxidants, acid deposition, persistent organic pollutants (POPs), polycyclic aromatic hydrocarbons (PAHs), radioactive nuclides, metals, and $PM_{2.5}$ (WHO, 2001; Cohen et al., 2004; Burns et al., 2011). The effect and effectiveness of aerosols on climate and pollution highly depends on their physical (size, shape, etc.) and chemical (hygroscopicity, refractive index, etc.) properties. The removal rates of aerosols, which increase earth surface contamination and decrease air concentrations, depend highly on these properties. The nonlinear degradation effect (or positive feedback) of surface and air pollution (surface concentration of aerosols increases due to their enhanced atmospheric stability) also depends on these properties. Because these aerosol properties substantially vary in time and space and the properties change during transport, it is essential to accurately simulate aerosol physical and chemical processes for both climate and pollution modeling.

In addition, the aerosol mixing state impacts the accuracy of the estimates of the bulk physical and chemical properties of aerosols (and thus their environmental impacts). Aerosols are more externally mixed near the emission source regions: aerosols from different emission sources exist discretely in the same air mass and in the same mode. Aerosols become more internally mixed over downwind regions: aerosols from different sources coagulate, and/or gas absorption and adsorption change the chemical composition of the aerosols. To express the high complexity of the aerosol mixing state and the changes in the mixing state from external to internal states during transport, a variety of aerosol representation methods have been used in atmospheric models. In this context, a category approach (e.g., Jacobson, 2002) has been developed and





used in both climate models (e.g., Vignati et al., 2004; Pringle et al., 2010; Aquila et al., 2011; Zhang et al., 2012; Liu et al., 2012, 2016) and chemical transport models (e.g., Riemer et al., 2003; Vogel et al., 2009; Kajino and Kondo, 2011; Kajino et al., 2012a; Kajino et al., 2019a, this study). In terms of black carbon (BC), mixing state resolving models have been developed (Oshima et al., 2009a; Ching et al., 2016a; Matsui, 2017) to explicitly treat the hydrophobic to hygroscopic

changes in BC-containing particles and increases in light absorption amplification and in-cloud scavenging efficiency through condensational growth during transport. The ultimate representation of the mixing state is a single particle resolving model for expressing a continuous mixing state (or an intermediate state of aerosol mixing) (Riemer et al., 2009; Zaveri et al., 2010), which has been implemented in a 1-D framework (Curtis et al., 2017); 3-D implementation is currently ongoing.

Nevertheless, computational resources are limited, and there are many important processes to consider in addition to

aerosol representations. Thus, it is essential to incorporate sufficiently efficient but realistic aerosol representations in coupled meteorology-chemistry models, with an awareness of the impacts of the aerosol mixing state on the bulk properties of populations of aerosols, as assessed by a series of single particle-based studies (e.g., Riemer et al., 2009; Zaveri et al., 2010; Ching et al., 2012; 2016b; 2017; 2018).

From the context mentioned above, in the current study, three options for aerosol representations, namely, 5-

category nonequilibrium, 3-category nonequilibrium and bulk equilibrium representations, are implemented in a three-dimensional regional-scale meteorology-chemistry model (Kajino et al., 2019a) and intercompared in this study. The three aerosol representations developed for the three respective purposes of regional climate, air quality, and operational forecasting. The 3-category method (Aitken mode, accumulation mode, and coarse mode) is a global standard method that has been widely used in various regional-scale chemical transport models (e.g., Grell et al., 2005; Byun and Schere, 2006).

The 5-category method includes two additional modes: dividing the accumulation mode into soot-free and soot-containing particles to consider differences between light-absorbing and nonlight-absorbing aerosols and dividing the coarse mode into dust (light-absorbing and hydrophobic) and sea salt (nonlight-absorbing and hygroscopic) particles. The bulk equilibrium method assumes instantaneous gas-aerosol partitioning and does not consider changes in the size distribution due to new particle formation, condensation, and coagulation for computational efficiency. The descriptions of the meteorological and

chemical models and their coupling procedures are briefly given in Sect. 2, with the details given in Supplement 1. The descriptions of the three aerosol representation options, the highlight of this study, are given in Sect. 3, followed by the descriptions of the simulation settings in Sect. 4. The model evaluations using the chemical, physical, and optical observation data are performed in Sect. 5. The differences in the model performances due to the selection of the three aerosol representation options are presented and discussed in Sect. 6. Conclusions are summarized in Sect. 7.

**2 NHM-Chem: a regional-scale meteorology-chemistry model**

NHM-Chem (v1.0) (Kajino et al., 2019a) is a chemical transport model (CTM) coupled with the Japan Meteorological Agency (JMA)'s nonhydrostatic model (NHM), either offline or online. The chemistry-to-meteorology feedback process has



not yet been included and will be implemented in the near future. The online version of NHM-Chem is currently a so-called 1-way (meteorology to chemistry) online coupled model. In this study, as the first step, the simulation results obtained with the offline coupled version were presented in this paper.

NHM solves fully compressible Navier-Stokes dynamics equations, considering major physical processes, such as
atmospheric radiation, cloud microphysical processes, and planetary boundary layer, surface layer, and land surface processes (Saito et al., 2006; 2007). NHM has been developed and used for the operational weather forecasting of JMA as well as for research purposes. The research and development history of this approach and its use in ongoing projects were thoroughly reviewed by Saito (2012). Currently, the next-generation nonhydrostatic model ASUCA (Asuca is a System based on a Unified Concept for Atmosphere; JMA, 2014; Aranami et al., 2015) is operational, and the NHM is used for
research purposes only.

The schemes of the CTM and database available for NHM-Chem are summarized in Table S1. NHM-Chem was briefly described in Kajino et al. (2019a). It is described in detail in Supplement 1 that NHM-Chem considers major tropospheric chemical, dynamical, and microphysical processes, such as advection (Walcek and Aleksic, 1998), turbulent diffusion, homogeneous and heterogeneous photochemistry (Carter, 2000; Jacob 2000; Edney et al., 2007), liquid-phase
chemistry (Walcek and Taylor, 1986; Carlton et al., 2007), new particle formation, condensation, coagulation, and dry deposition (Kajino et al., 2012a), fog deposition (Katata et al., 2015), in-cloud scavenging of aerosols (CCN and ice nuclei (IN) activation and subsequent cloud microphysical processes), below-cloud scavenging of aerosols (collection by settling hydrometeors), and wet deposition of gases (dissolution to cloud and rain droplets) (Kajino et al., 2012a; Kajino et al., 2019a), and subgrid-scale convective transport and deposition (Pleim and Chang, 1992).

The coupling between meteorology and chemistry is illustrated in Fig. S1-1. The offline coupled version has been used for various purposes, such as the simulation of dust vortices in the Taklimakan Desert (Yumimoto et al., 2019), the simulation of the dispersion and deposition of radionuclides due to the Fukushima nuclear accident (Kajino et al., 2019b; Sekiyama and Kajino, 2020), simulation of lower-tropospheric ozone in East Asia (Kajino et al., 2019c), and the simulation of transition metals in East Asia (Kajino et al., 2020). Other meteorological models, such as the Weather Research and
Forecast model (WRF; Skamarock et al., 2008), ASUCA (JMA, 2014; Aranami et al., 2015), and Scalable Computing for Advanced Library and Environment (SCALE; Nishizawa et al., 2015, Sato et al., 2015), can be used for offline coupled simulations (Fig. S1-1). The offline NHM-Chem with WRF was used for a multi-CTM intercomparison study in Asia (Li et al., 2019; Chen et al., 2019; Itahashi et al., 2020; Kong et al., 2020; Tan et al., 2020; Ge et al., 2020) and a multimeteorological model study of the Fukushima nuclear accident (Kajino et al., 2019b). The offline model with SCALE
was used for a multi-CTM intercomparison study for the Fukushima nuclear accident (Sato et al., 2020) and that with ASUCA is currently being used by JMA to produce an operational forecast for photochemical smog bulletins (http://www.jma.go.jp/jma/kishou/know/kurashi/smog.html; last access: 18 June 2020, in Japanese).



## 3 Aerosol representations

There are currently three options for the representations of aerosol categories (or types), namely, the 5-category nonequilibrium method, the 3-category nonequilibrium method, and the bulk equilibrium method (Fig. 1), to simulate general air quality issues (related to factors such as photochemical oxidants, $PM_{2.5}$, mineral dust, and acid deposition).

Further simplified modules were developed to predict the transport of radionuclides (Kajino et al., 2019b) and transition metals (Kajino et al., 2020), which do not simulate the photochemistry and some aerosol dynamics processes, such as new particle formation, condensation, and coagulation, and thus are not assessed in this study. The prognostic variables of the aerosol attributes of these three methods are listed in Table 1. Each elementary process is depicted by a colored arrow in Fig. 1. Here, the term "bulk equilibrium" indicates that the gas-aerosol partitioning of inorganic semivolatile components, such as

$NH_3$-$NH_4^+$ $HNO_3$-$NO_3^-$, and $HCl$-$Cl^-$, is instantaneously attained at the equilibrium state. On the other hand, in the nonequilibrium methods, mass transfer coefficients are calculated for each category based on its size distribution, and condensation and evaporation are driven by the mass transfer coefficients multiplied by the differences between the equilibrium state and the current state of each category (see Eqs. 13-16 of Kajino et al., 2012a).

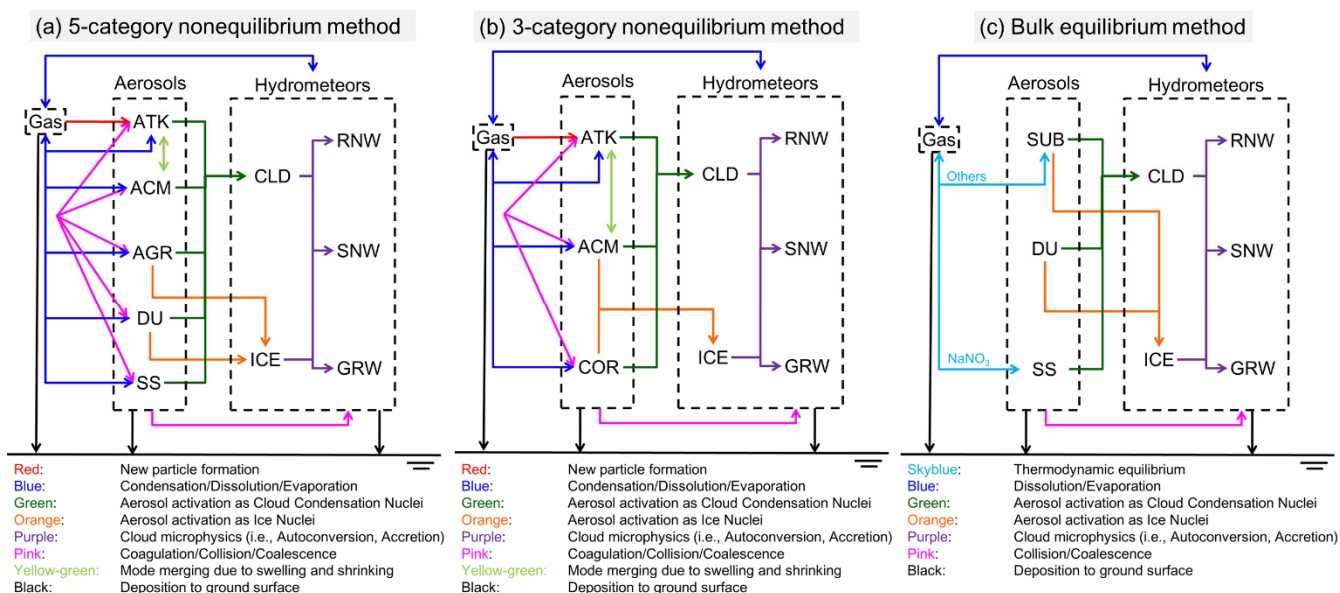

Figure 1: Schematic illustrations of gas-aerosol-cloud dynamic processes considered in (a) 5-category, (b) 3-category (global standard), and (c) bulk equilibrium methods. ATK (Aitken mode), ACM (soot-free accumulation mode, 5-category; accumulation mode, 3-category), AGR (accumulation mode, mixed with soot aggregate) DU (mixture with dust; both

mineral and anthropogenic dust), SS (mixture with sea salt), COR (coarse mode), SUB (submicron), CLD (cloud), ICE (cloud ice), RNW (rain), SNW (snow), GRW (graupel). This figure is same as Fig. 1 of Kajino et al. (2019a).





Table 1: Prognostic variables of aerosol-related attributes

| Aerosol options (number of tracers) | Category name | Description of category | Physical properties | | Chemical compositions | | | | | | | | | |
|---|---|---|---|---|---|---|---|---|---|---|---|---|---|---|
| 5-category (50) | ATK | Aitken mode | $M_0$[a] | $M_2$[b] | | | OM[c] | | | $SO_4^{2-}$[d] | $NO_3^-$ | $NH_4^+$ | $Cl^-$ | $H_2O$ |
| | ACM | Accumulation mode, soot-free | $M_0$ | $M_2$ | UID[e] | | OM | | | $SO_4^{2-}$ | $NO_3^-$ | $NH_4^+$ | $Cl^-$ | $H_2O$ |
| | AGR | Accumulation mode, mixed with soot aggregate | $M_0$ | $M_2$ | UID | BC | OM | | | $SO_4^{2-}$ | $NO_3^-$ | $NH_4^+$ | $Cl^-$ | $H_2O$ |
| | DU | dust | $M_0$ | $M_2$ | UID | BC | OM | MD[f] | | $SO_4^{2-}$ | $NO_3^-$ | $NH_4^+$ | $Cl^-$ | $H_2O$ |
| | SS | sea salt | $M_0$ | $M_2$ | UID | BC | OM | MD | NS[g] | $SO_4^{2-}$ | $NO_3^-$ | $NH_4^+$ | $Cl^-$ | $H_2O$ |
| 3-category (38) | ATK | Aitken mode | $M_0$ | $M_2$ | | | OM | | | $SO_4^{2-}$ | $NO_3^-$ | $NH_4^+$ | $Cl^-$ | $H_2O$ |
| | ACM | Accumulation mode | $M_0$ | $M_2$ | UID | BC | OM | | | $SO_4^{2-}$ | $NO_3^-$ | $NH_4^+$ | $Cl^-$ | $H_2O$ |
| | COR | Coarse mode | $M_0$ | $M_2$ | UID | BC | OM | MD | NS | $SO_4^{2-}$ | $NO_3^-$ | $NH_4^+$ | $Cl^-$ | $H_2O$ |
| Bulk (20) | SUB | Bulk submicron | $M_0$ | $M_2$ | UID | BC | OM | | | $SO_4^{2-}$ | $NO_3^-$ | $NH_4^+$ | $Cl^-$ | $H_2O$ |
| | DU[h] | Dust | $M_0$ | $M_2$ | UID | | | MD | | | | | | |
| | SS[i] | Sea salt | $M_0$ | $M_2$ | | | | | NS | | $NO_3^-$ | | $Cl^-$ | $H_2O$ |

[a] 0th moment, equal to number concentration
[b] 2nd moment, proportional to surface area concentration
[c] Total (primary plus secondary) organic mass, speciation of secondary organics is treated as bulk (gas+aerosol) species. This treatment is feasible under the assumption that the molecular speeds and diffusivities of all organics are the same.
[d] Non-sea salt $SO_4^{2-}$ only.
[e] Unidentified components, such as anthropogenic dust.
[f] mineral dust mass (natural).
[g] Nonvolatile components of pure sea salt mass (chloride is treated separately as it is volatile).
[h] Assumed to be chemically inert
[i] Only reacted with $HNO_3$ gas assuming instantaneous thermodynamic equilibrium

       The 3-category method (ATK: Aitken mode, ACM: accumulation mode, COR: coarse mode) is a global standard method widely used in regional-scale CTMs, such as WRF-Chem and CMAQ (WRF-Chem and CMAQ provide other aerosol schemes, such as sectional methods, in addition to the standard 3 modal method, e.g., Chapman et al., 2009; Zhang et al., 2004; 2010). These models were originally developed for the purpose of air quality prediction, focusing on factors such as acid deposition, photochemical oxidants, and $PM_{2.5}$ (Byun and Schere, 2006; Grell et al., 2005); however, they have recently been used to perform regional climate predictions considering aerosol-cloud-radiation interaction processes (Chapman et al., 2009; Wong et al., 2012; Kajino et al., 2017). However, the 3-category method does not consider the two aspects that are critically important in climate simulations and are thus often resolved by climate models (e.g., Vignati et al., 2004; Pringle et al., 2010; Aquila et al., 2011; Zhang et al., 2012; Liu et al., 2012, 2016): (1) the differences between light-





absorbing and nonlight-absorbing aerosols and (2) the differences between mineral dust and sea salt particles. To consider these two aspects, the 5-category method is implemented by dividing the accumulation mode (ACM of the 3-category method) into the ACM (soot-free accumulation mode) and AGR (accumulation mode mixed with soot aggregate) categories and by dividing the coarse mode (COR of the 3-category method) into the DU (mixture with dust; i.e., both mineral dust and anthropogenic dust) and SS (mixture with sea salt) categories.

BC is a strong light-absorbing agent and thus alters the climate by heating the air (Bond et al., 2013 and references therein). BC is usually hydrophobic when emitted and externally mixed with other aerosols to become more hygroscopic, forming an internal mixture with other components, including water-soluble inorganics, such as sulfate ($SO_4^{2-}$), nitrate ($NO_3^-$), and ammonium ($NH_4^+$), due to condensation and coagulation during transport (Oshima and Koike, 2013). The in-cloud

scavenging of BC, which is the major removal process of BC (Kondo et al., 2011; Oshima et al., 2012), highly depends on the mixing state of BC (Ching et al., 2012; 2016b; 2018). Certainly, the 5-category method cannot resolve the mixing state of BC; however, it can simulate the bulk amounts condensed onto BC vs. non-BC particles, whereas the 3-category method cannot. Thus, the 3-category method will overestimate the condensational growth of BC and therefore overestimate its hygroscopicity, in-cloud scavenging, and subsequent heterogeneous ice nucleation (i.e., immersion and condensation

freezing). All of the $SO_4^{2-}$ and $NO_3^-$ condense onto BC in the 3-category method, although major proportions of $SO_4^{2-}$ and $NO_3^-$ are mixed with non-BC particles in reality (Miyakawa et al., 2014). The degree of overestimation caused by the 3-category method relative to the 5-category method is presented later in Sect. 6.4.

The 3-category method assumes a completely internal mixture of dust and sea salt when the two particles coexist in the same grid box, although the chemical and physical properties of these two particles are very different. In Japan, massive

transport events of mineral dust originating from Chinese arid regions, such as the Gobi and Taklimakan Deserts, often occur in the spring associated with the cold front of migrating anticyclones. Zhang and Iwasaka (2004) found that 10-20% of number fractions of coarse mode particles were mixtures of mineral dust and sea salt, with similar mass fractions, during dust events in the spring of 1996. They also found only 5-15% of pure dust particles and as much as 60-80% of mixtures of mineral dust and sea salt, including dust mixed with a small amount of sea salt and sea salt mixed with a small amount of

dust. Mineral dust is hydrophobic and nonspherical when emitted and thus shows high IN activity, is light-absorbing, becomes hygroscopic during transport, and invigorates heterogeneous reactions (Kameda et al., 2016). On the other hand, sea salt is highly hygroscopic and is thus an efficient CCN. The size distributions of mineral dust and sea salt are also very different, so in this 3-category method, when the two particles coexist in the same grid box, their combined size distributions and hygroscopicity become unrealistic (i.e., the intermediate values are different from those of both particle types, as shown

later in Sect. 6). This 3-category representation would be adequate for predicting air pollution over a continent, such as Europe or America. It would also be safe for global climate simulations because on average, the major global proportions of dust and sea salt exist separately in different locations, namely, over the continents and oceans, respectively. However, for cases of long-range transport over the Asian continent to Japan, mineral dust and anthropogenic pollutants travel in the sea salt-rich boundary layer over the ocean. As discussed in Sect. 6, the 3-category method is thus not suitable for air quality and



regional climate predictions in East Asia. Consequently, the 5-category method is regarded as the standard method of NHM-Chem.

The 5-category and 3-category methods fully solve for aerosol microphysical processes by using the triple-moment modal method presented in the previous section (see red, blue, and pink arrows in Figs. 1a and 1b). Even though a modal approach, which is basically computationally efficient compared to a sectional approach, is used, it is still time-consuming because NHM-Chem explicitly models the homogeneous nucleation of sulfuric acid gas to produce 1 nm particles and their subsequent growth due to condensation and coagulation. In terms of the operational forecasts of air quality, such as photochemical oxidants, mineral dust, and $PM_{2.5}$, the surface mass concentration is of primary importance, and the physical and chemical properties of aerosols are of secondary importance. Assessing the uncertainty in the initial and boundary conditions is critical for obtaining accurate predictions of mass concentrations; thus, data assimilation is required. For the purpose of obtaining an operational forecast, together with data assimilation, the computationally efficient bulk equilibrium method is developed, which does not solve aerosol microphysical processes, such as new particle formation, condensation, and coagulation (compare the sky blue arrows in Fig. 1c with the red, blue, and pink arrows in Figs. 1a and 1b). Aerosols are categorized into three categories, namely, SUB (submicron), DU (dust), and SS (sea salt). The total dry mass of SUB can be used for the prediction of $PM_{2.5}$ and the mineral dust mass (denoted as MD) in the DU category. As mentioned earlier in this subsection, thermodynamic equilibrium is assumed for the inorganic components, and all of the aerosol-phase components are assumed to be categorized into the SUB category, except for $NaNO_3$, which is mixed with sea salt in the SS category. Most of the $SO_4^{2-}$ and $NH_4^+$ exist at the submicron size range, whereas $NO_3^-$ in Japan is usually partitioned into both submicron and supermicron (or coarse mode) size ranges because the latter is internally mixed with sea salt or mineral dust particles (Kajino et al., 2012b; Kaneyasu et al., 2012; Kajino et al., 2013; Uno et al., 2016; Uno et al., 2017). Without considering the coarse mode $NO_3^-$, the simulated $PM_{2.5}$ in East Asia will be significantly overestimated due to the overestimated production of $NH_4NO_3$. Certainly, some of the coarse mode $NO_3^-$ should be mixed with dust, but the DU category is assumed to be inert in the bulk equilibrium method for the purposes of computational efficiency.

As listed in Table 1, the numbers of aerosol attributes for the 5-category, 3-category, and bulk equilibrium methods are 50, 38, and 20, respectively. In the triple-moment method, the 0th moment (number) and 2nd moment (proportional to the surface area) are the prognostic (transported) variables, and the 3rd moment (proportional to the volume) is diagnosed based on the mass concentrations of each component and their prescribed densities (i.e., 2.0, 1.77, 2.6, 2.2, 1.83, and 1.0 g cm$^{-3}$ for UID (unidentified mass, or anthropogenic dust), carbonaceous mass (BC and OM), mineral dust (MD), sea salt (nonvolatile components of sea salt (NS) and $Cl^-$), inorganics ($SO_4^{2-}$, $NO_3^-$, and $NH_4^+$), and water ($H_2O$), respectively. Here, note that the NS indicates the nonvolatile mass of pure sea salt particles, namely, the dry sea salt mass minus the $Cl^-$ mass.





Table 2: Rules for the transfer of moments and chemical mass concentrations from one category to another due to intercategory coagulation and mode merging

| Aerosol options | Rule |
|---|---|
| 5-category | *Intercategory coagulation* |
| | ATK + ACM → ACM |
| | ATK + AGR → AGR |
| | ATK + DU → DU |
| | ATK + SS → SS |
| | ACM + AGR → AGR |
| | ACM + DU → DU |
| | ACM + SS → SS |
| | AGR + DU → DU |
| | AGR + SS → SS |
| | DU + SS → SS |
| | *Mode merging[a]* |
| | ATK → ACM |
| 3-category | *Intercategory coagulation* |
| | ATK + ACM → ACM |
| | ATK + COR → COR |
| | ACM + COR → COR |
| | *Mode merging[a]* |
| | ATK → ACM |
| Bulk | n.a. |

[a]Transfer of moments and mass greater than 40 nm due to condensation or self-coagulation growth

The rules for the transfer of moments and mass from one category to another due to intercategory coagulation and mode merging are listed in Table 2. The intercategory coagulation rules of "from smaller to larger category" are used for the 3-category and 5-category methods; additionally, the coagulation rules of "from hydrophobic to hygroscopic category" and "from nonlight-absorbing to light-absorbing category" are used for the 5-category method. The mode merging represents the transfer of moments and mass greater than the threshold diameter of 40 nm due to condensation or self-coagulation growth. During its emission, the UID only exists in ACM and DU in the 5-category method, in ACM and COR in the 3-category method, and in SUB and DU in the bulk method. Due to intercategory coagulation, the UID is also found in AGR and SS in the 5-category method. Freshly emitted BC only exists in AGR, ACM, and SUB in the 5-category, 3-category, and bulk methods, respectively. The intercategory coagulation transfers BC to the DU and SS categories in the 5-category method and to COR in the 3-category method. OM exists in ATK, ACM, and AGR in the 5-category method, in ATK and ACM in the 3-category method, and in the SUB category in the bulk method during its emission. OM is moved to the DU and SS categories in the 5-category method and to the COR category in the 3-category method. The MD mass only exists in the DU category for the 5-category and bulk methods, and it exists in the COR for the 3-category method. Transfer of the MD mass to the SS category is predicted only with the 5-category method. All of the secondary and semivolatile components, such as





OM, $SO_4^{2-}$, $NO_3^-$, $NH_4^+$, and $Cl^-$, can exist in all of the categories in the 5-category and 3-category methods. They can exist only in SUB in the bulk method, except for $NO_3^-$ and $Cl^-$, which can mix with sea salt, as mentioned earlier ($Cl^-$ also exists in the SS category during emission). All of the aerosol categories can be hygroscopic and thus contain $H_2O$, except for the DU category of the bulk method. As mentioned earlier, crustal cations play an essential role in fixing acidic gases to the aerosol

phase. In all three methods, the compositions of the sea salt and dust particles are defined after Song and Carmichael (2001) as follows: the MD mass contains 6.8, 1.6, 0.91, and 0.78 wt% $Ca^{2+}$, $Na^+$, $K^+$, and $Mg^{2+}$, respectively, and the NS mass contains 68.1, 17.1, 8.21, 2.58, and 2.45 wt% $Na^+$, $SO_4^{2-}$, $Mg^{2+}$, $Ca^{2+}$, and $K^+$, respectively. It is assumed that the UID mass does not contain any cations and is thus hydrophobic and inert.

In terms of aerosol water uptake, nonequilibrium treatment is critically important, especially for higher relative

humidity conditions and coarse mode particles, because the instantaneous equilibrium assumption can cause unrealistic overestimates of aerosol diameter. For example, the equilibrium diameter of an aerosol with a dry diameter of 1.0 μm and a hygroscopicity $\kappa$ value of 1.0 reaches 21.2 μm at a relative humidity RH = 100% (see $\kappa$-Köhler theory; Petters and Kreidenweis, 2007), which is equal to or greater than the initial sizes of cloud droplets. Under actual atmospheric conditions, aerosols cannot reach such a large equilibrium diameter, either because it takes too long time to reach (for low aerosol

populations) or because aerosols remove too much water vapor from the air, such that the relative humidity becomes substantially lower (for high enough aerosol populations). Therefore, in the 3-category and 5-category nonequilibrium methods, aerosol water uptake is also calculated in a nonequilibrium manner: the mass transfer of water vapor is driven by the differences between the current water content and equilibrium water content, which is derived using the ZSR method (Fountoukis and Nenes, 2007). Note that the RH of air is assumed to remain unchanged due to water uptake by aerosols. In

the bulk equilibrium method, instantaneous equilibrium is assumed for the aerosol water content, but a maximum threshold of an ambient RH of 98% is used to avoid the abovementioned unrealistically large aerosol size.

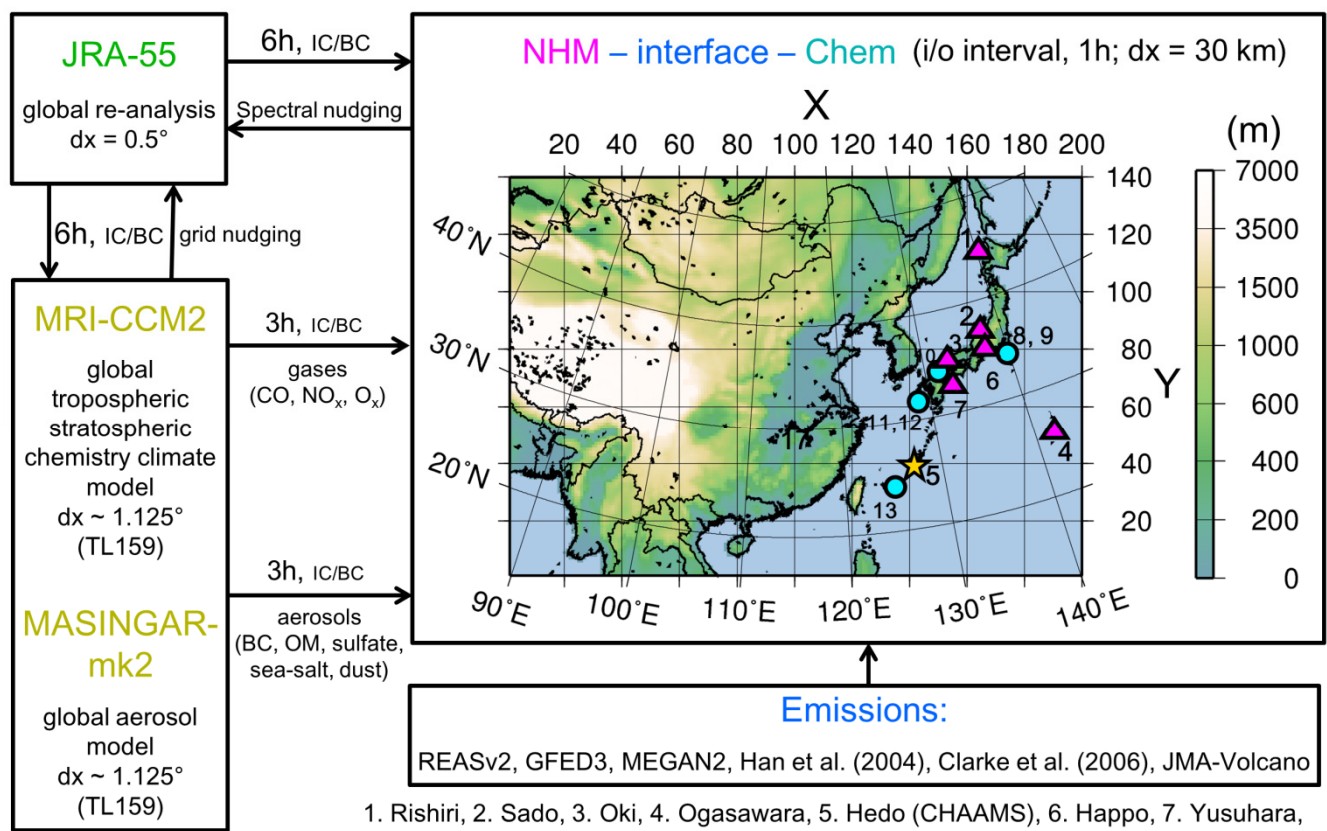

1. Rishiri, 2. Sado, 3. Oki, 4. Ogasawara, 5. Hedo (CHAAMS), 6. Happo, 7. Yusuhara,
8. Chiba, 9. Tsukuba, 10. Matsue, 11. Kasuga, 12. Fukue, 13. Miyako

Figure 2: Current simulation settings, model domain showing terrestrial elevation (m) and observation sites. The pink triangles numbered 1-7 show the Japanese EANET sites, except site 5. Hedo (CHAAMS) is marked as the yellow start, where the SKYNET and AD-Net Hedo sites are located on the same premises. The blue circles numbered 8-13 indicate the SKYNET or AD-Net sites in Japan.

## 4 Experimental setup

### 4.1 Model domain, simulation period, and boundary conditions

The simulation settings such as the model domain and boundary conditions used in this study are presented in this section. They were the same as Kajino et al. (2019a) and thus are briefly described in this section; detailed descriptions, especially of the aerosols, are provided in Sect. 4.2. Figure 2 shows a flowchart of the simulations, including the simulation domain, which covers East Asia with $200 \times 140$ horizontal grid cells with a grid resolution of $\Delta x = 30$ km. The numbers of vertical grid cells in the NHM and CTM are 38 (reaching up to 22,055 m above sea level (ASL)) and 40 (reaching up to 18,000 m ASL), respectively, with terrain-following coordinates. The input/output time interval of the offline coupled NHM-Chem was 1 hour. The analysis period was the entire year of 2006, but two half-year CTM simulations were conducted and then



combined with a spin-up period of 5 days. Climatological data were used for the initial condition of the land surface model (Simple Biosphere Model, SiB) of the NHM. If the simulation started in winter, the initial climatological snow coverage could be very different from that at the actual simulation time; thus, a longer spin-up period would be required. To avoid this situation, July was selected as the initial simulation time because over a year, July corresponds to the minimum snow coverage over East Asia. Therefore, two NHM simulations were performed, from 26 June 2005 to 1 July 2006 and from 26 June 2006 to 1 January 2007. Then, the two CTM simulations were conducted from 27 December 2005 to 1 July 2006 and from 26 June 2006 to 1 January 2007. Then, the two simulation results from 1 January 2006 to 1 July 2006 and from 1 July 2006 to 1 January 2007 were combined and used in the following analysis.

The 6-hourly JRA-55 global reanalysis (Kobayashi et al., 2015) data were used to set the initial and boundary conditions of the NHM. Spectral nudging was applied to constrain the simulated meteorological field toward the global analysis, above a height of 7 km for large-scale wave components (wavelength > 1,000 km) of horizontal momentum and potential temperature, with a weighting factor of 0.06 (Nakano et al., 2012). Data assimilation was not applied to the chemical fields in this study.

The 3-hourly lateral and upper boundary concentrations of gases and aerosols were obtained from the simulation results of the global models of MRI-CCM2 (Deushi and Shibata, 2011) (TL159, $\Delta x \sim 1.125°$) and MASINGAR mk-2 (Tanaka et al., 2003; Tanaka and Ogi, 2017; Yumimoto et al., 2017) (TL159, $\Delta x \sim 1.125°$), respectively. Because the lumped mechanisms of nonmethane volatile organic compounds (NMVOCs) and aerosol representations of NHM-Chem are different from those of the two global models, only $NO_x$, $O_3$, and CO were taken from the MRI-CCM2 (in other words, the lateral boundaries should be far enough that the peroxyacetyl nitrate (PAN) and long-lived NMVOC fluxes from the boundaries are less influential on the concentrations over the targeted sites of the simulation), and the total masses of BC, OC, $SO_4^{2-}$, mineral dust, and sea salt were taken from MASINGAR mk-2 (although this model employs sectional methods for mineral dust and sea salt). An OM to OC ratio of 1.8 and a 100% existence of $SO_4^{2-}$ as $(NH_4)_2SO_4$ were assumed in the boundary concentrations. The mineral dust and sea salt were treated as being externally mixed, whereas all of the BC, OM, and $(NH_4)_2SO_4$ were included in the ACM in the 3-category method and in the SUB category in the bulk equilibrium method. For the 5-category method, we assumed that 80% of the BC, OM and $(NH_4)_2SO_4$ were attributed to ACM and that the other aerosols were attributed to AGR.

## 4.2 Emission data sets

We used REASv2 (Kurokawa et al., 2013) with a 0.25°×0.25° resolution for the anthropogenic emissions of $NO_x$, $SO_2$, $NH_3$, NMVOC, BC, POC, $PM_{2.5}$ and $PM_{10}$ with monthly variations. There is no monthly variation in $NH_3$. Because REASv2 does not provide hourly and vertical profiles of emissions, those of Li et al. (2017) were applied to each sector, i.e., power, industry, domestic, transport, aviation, and large point sources. We used the monthly Global Fire Emissions Database (GFED3; Giglio et al., 2010) with a 0.5°×0.5° resolution for open biomass burning emissions ($NO_x$, $SO_2$, NMVOCs, BC, and POC) and the Model of Emissions of Gases and Aerosols from Nature (MEGAN2; Guenther et al., 2006) for biogenic





emissions of isoprene, terpenes, methanol, and NO. We applied the monthly mean values of GFED3 without temporal variations. The temporally varying biogenic emission flux was calculated by Eq. 1 of Guenther et al. (2006) using the surface solar radiation and surface air temperature, as simulated by NHM. Biogenic emissions were allocated to the bottom layer. The open biomass burning emissions were uniformly allocated from the bottom layer up to 1,000 m above ground level

(AGL). The lightning $NO_x$ emission was not considered in the current simulation.

We made the following assumptions for the ratios of lumped species in the anthropogenic and biomass burning emissions. REASv2 provided speciation information about NMVOC, which was redistributed to the NMVOC speciation of SAPRC99. Because GFEDv3 does not provide NMVOC speciation information, that of Woo et al. (2003) was applied and allocated to the SAPRC99 speciation. When a SAPRC99 speciation was finer than that of REASv2 or GFEDv3, the

speciated NMVOC of REASv2 or GFEDv3 was equally distributed to that of SAPRC99 with equal mixing ratios. The OM to OC ratio was set to 1.8 for both emissions. The $NO_x$ emissions were partitioned as 90% NO and 10% $NO_2$. Five percent of $SO_2$ emissions were regarded as $SO_4^{2-}$. The UID emissions of the fine and coarse modes were defined as $PM_{2.5}$ minus the BC minus the POM (primary organic mass; $1.8 \times POC$) and $PM_{10}$ minus $PM_{2.5}$, respectively. The $SO_4^{2-}$ emissions were partitioned as 10% ATK and 90% ACM for the 5-category and 3-category methods. Of the POM emissions, 5%, 10%, and

85% were distributed to ATK, AGR, and ACM in the 5-category method, whereas 5% and 95% of these emissions were distributed to ATM and ACM in the 3-category method, respectively.

Hourly volcanic $SO_2$ emissions in Japan were developed in this study. These emissions were assumed to comprise 100% $SO_2$, and no volcanic $SO_4^{2-}$ emissions were considered. JMA regularly monitors the $SO_2$ emission fluxes and smoke heights of the six major volcanoes in Japan. Since the observation data are sporadic over time and the observation

frequencies vary depending on the volcanoes and periods considered, cubic spline interpolation was applied over time to obtain hourly data, as done by Kajino et al. (2004). The details of this approach are described in Supplement 2.

The methods of Han et al. (2004) and Clarke et al. (2006) were used to predict the emissions of mineral dust and sea salt particles as functions of the friction velocity and 10-m wind speed, respectively. In addition to the formula of Han et al. (2004), simulated snow coverage was applied to reduce the emission flux. Flux adjustments were made for the mineral dust

and sea salt emissions by using the observed non-sea-salt (nss) $Ca^{2+}$ (defined as $[Ca^{2+}] - 0.038 \times [Na^+]$ ($\mu g/m^2$ or $\mu g/m^3$)) and $PM_{10}$ values for the mineral dust and the observed $Na^+$ and $PM_{10}$ values for the sea salt obtained at the seven Japanese Acid Deposition Monitoring Network in East Asia (EANET) stations. The adjustment ratios of the dust and sea salt emission fluxes in the simulations were 0.5 and 0.3, respectively.



Table 3: Symbols used for aerosol size parameters and their relationships

| Symbol | Description | Relationship |
|---|---|---|
| $\sigma_g$ | Geometric standard deviation | |
| $\rho_{p,dry}$ | Particle dry density (g/cm$^3$) | |
| $\rho_p$ | Particle density (g/cm$^3$)[a] | |
| $D_{g,n,dry}$ | Number-equivalent geometric mean dry diameter | |
| $D_{g,m,dry}$ | Mass-equivalent geometric mean dry diameter | $= D_{g,n,dry} \times \exp\left(3 \times \ln(\sigma_g)^2\right)$ |
| $D_{g,n,aero,dry}$ | Aerodynamic $D_{g,n,dry}$ | $= D_{g,n,dry} \times \sqrt{\rho_{p,dry}}$ [b] |
| $D_{g,m,aero,dry}$ | Aerodynamic $D_{g,m,dry}$ | $= D_{g,m,dry} \times \sqrt{\rho_{p,dry}}$ [b] |
| $D_{g,n}$ | Number-equivalent geometric mean diameter[a] | |
| $D_{g,m}$ | Mass-equivalent geometric mean diameter[a] | |
| $D_{g,n,aero}$ | Aerodynamic $D_{g,n}$ | $= D_{g,n} \times \sqrt{\rho_p}$ [b] |
| $D_{g,m,aero}$ | Aerodynamic $D_{g,m}$ | $= D_{g,m} \times \sqrt{\rho_p}$ [b] |

[a]When wet, or ambient
[b]For spheres

## 4.2 Aerosol size parameters in emission and boundary conditions

The symbols used for the aerosol parameters used in this study are summarized in Table 3. Aerosol sizes should be carefully defined because both their physical and optical measures significantly vary depending on whether their diameters are number-equivalent or mass-equivalent, wet or dry, and actual or aerodynamic. The width of the distribution $\sigma_g$, hygroscopicity $\kappa$, and aerosol density $\rho_p$ are also key parameters because they affect the magnitudes of their physical and optical measures. Certainly, shape factors can also significantly affect these measures, as well as aerosol microphysical processes; however, all of the modeled particles are assumed to be spherical in the current version of NHM-Chem, as they are in most other CTMs.

Both Han et al. (2004) and Clarke et al. (2006) provided size-resolved emission fluxes, whereas this model assumes a log-normal size distribution. The values of the number equivalent dry $D_g$ ($D_{g,n,dry}$) of the mineral dust and sea salt emissions were obtained by assuming the prescribed $\sigma_g$ as 2.0. The $D_{g,n,dry}$ values of the mineral dust were 0.66, 1.22, and 1.22 μm for the land use category (LUC) of desert, loess, and grass, respectively. For a reference, $D_{g,n,dry}$ values of 0.66 and 1.22 μm correspond to aerodynamic $D_{g,n,dry}$ ($D_{g,n,aero,dry}$) values of 1.06 and 1.96 μm, mass-equivalent dry aerodynamic $D_g$ ($D_{g,m,aero,dry}$) values of 4.50 and 8.31 μm, and PM$_{2.5}$/PM$_{10}$ ratios of 0.23 and 0.068, respectively. Note that here, the simulated PM$_x$ was derived as a proportion of the dry mass in which the aerodynamic ambient (wet) diameter was smaller than $x$ μm.





The simulated $PM_x$ was calculated using the error function (erf) as follows:

$$PM_x = \sum_i \frac{M_i}{2} \, \text{erf}\left( \frac{\ln D_x - \ln D_{g,m,aero,i}}{\sqrt{2} \ln \sigma_{g,i}} \right),$$

(1)

where $i$ and $M_i$ indicate the category and dry mass of the category, respectively.

We made the following assumptions for the aerosol size distributions at the boundary conditions. Because Clarke et al. (2006) proposed trimodal size distributions, they needed to be combined into a single mode. Laboratory and field experiments indicated that the major proportions of the smallest mode (i.e., 10-132 nm) should be water-insoluble organic compounds (Facchini et al., 2008; Prather et al., 2013); thus, the smallest mode was not regarded as sea salt emissions in this model. The number and volume fluxes of the sea salt emission were obtained as the summations of those of the second-largest and largest modes. The surface area flux was then deduced based on the number, volume, and prescribed value of $\sigma_g =$ 2.0. The $D_{g,n,aero,dry}$ value, $D_{g,m,aero,dry}$ value, and $PM_{2.5}/PM_{10}$ ratio of the dry sea salt particles are 0.448 μm, 1.89 μm, and 0.66, respectively. Because sea salt particles are highly hygroscopic, at conditions of RH = 80% and 90%, the ambient diameter $D_{g,m,aero}$ values would be 2.39 and 2.88 μm, and the $PM_{2.5}/PM_{10}$ ratios would be 0.54 and 0.44, respectively, assuming a hygroscopicity value of $\kappa = 1.0$.

The prescribed size parameters were applied to the emissions of all of the species, except for the mineral dust and sea salt and are defined as follows: $(D_{g,n,aero,dry}, \sigma_g)$ = (0.01 μm, 1.3), (0.1 μm, 1.5), (0.1 μm, 1.5), and (2 μm, 1.8) for the ATK, ACM, AGR, and DU (applied to anthropogenic dust only, denoted as UID) categories of the 5-category method; (0.01 μm, 1.3), (0.1 μm, 1.5), and (2 μm, 1.8) for the ATK, ACM, and COR (applied to UID only) categories of the 3-category method; and (0.1 μm, 1.5) and (2 μm, 1.8) for the SUB and DU (applied to UID only) categories of the bulk equilibrium method, respectively.

Although MASINGAR mk-2 employs a sectional model to predict changes in aerosol size distributions, the prescribed size parameters were applied to the lateral and upper boundary concentrations of NHM-Chem and are defined as follows: $(D_{g,n,aero,dry}, \sigma_g)$ = (0.01 μm, 1.3), (0.1 μm, 1.5), (0.1 μm, 1.5), (2 μm, 1.8), and (1 μm, 1.8) for the ATK, ACM, AGR, DU and SS categories of the 5-category method; (0.01 μm, 1.3), (0.1 μm, 1.5), and (1 μm, 1.8) for the ATK, ACM, and COR (mineral dust and sea salt) categories of the 3-category method; and (0.1 μm, 1.5), (2 μm, 1.8), and (1 μm, 1.8) for the SUB, DU, and SS categories of the bulk method, respectively. Certainly, in the self-nesting simulations, where the boundary concentrations are taken from the outer domain simulation of NHM-Chem, the prescribed size parameters are not needed.



## 5 Observations

The observation data used for the evaluations of the simulation results are described in this section.

The observation sites used for the model validation are depicted in Fig. 2. The pink triangles indicate the Acid Deposition Monitoring Network in East Asia (EANET) stations (http://www.eanet.asia, last access: 18 June 2020). The blue

circles indicate the ground-based international remote sensing network dedicated to aerosol-cloud-radiation interaction studies (SKYNET) stations (Takamura et al., 2004; Nakajima et al., 2007; http://atmos3.cr.chiba-u.jp/skynet, last access: 18 June 2020) and/or the Asian dust and aerosol lidar observation network (AD-Net) stations (Sugimoto et al., 2008; Shimizu et al., 2016; http://www-lidar.nies.go.jp/AD-Net/, last access: 18 June 2020). At the Japanese EANET sites, the hourly surface concentrations of gases ($NO_x$, $SO_2$, and $O_3$) and the aerosol mass ($PM_{2.5}$ (only Oki and Rishiri for 2006) and $PM_{10}$), daily wet

depositions of inorganic compounds, and two or one week mean concentrations of gaseous and aerosol phases of inorganic compounds are monitored. At the SKYNET sites, sky radiometers (Prede Co. Ltd., Tokyo, Japan) are the main instruments. Aerosol optical thickness (AOT) and single scattering albedo (SSA) data in ultraviolet/visible/near-infrared regions and the Ångström exponent data are provided. We used AOT and SSA at 500 nm for the comparison. The AD-Net sites are equipped with a two-wavelength (1064 and 532 nm) polarization-sensitive (532 nm) Mie-scattering lidar system. We used the

extinction coefficients for pherical (referred to as dust) and spherical aerosols for the model evaluation, derived using the attenuated backscattering coefficients and the volume depolarization ratio (Sugimoto et al., 2003; Shimizu et al., 2004). The yellow star is the Hedo site, which belongs to both EANET and SKYNET, an air quality measurement supersite called CHAAMS (Cape Hedo Atmosphere and Aerosol Monitoring Station).

The OPC (ROYCO LAS236) was equipped at CHAAMS in 2006, which measures number concentrations of

aerosols with diameters larger than 0.3, 0.5, 1, 3, and 5 μm (Takamura et al., 2004). To compare the simulated number concentrations against the OPC data, a parameter $NC_x$ is defined, the proportion of the number concentration for which the aerodynamic ambient diameter is larger than $x$ μm, as follows:

$$NC_x = \sum_i \frac{N_i}{2} \left[ 1 - \mathrm{erf} \left( \frac{\ln D_x - \ln D_{g,n,aero,i}}{\sqrt{2} \ln \sigma_{g,i}} \right) \right], \tag{2}$$

where $i$ and $N_i$ indicate the category and total number of the category, respectively. The simulated $NC_{0.3}$ was compared against that observed by OPC.

In the current study, a model performance comparison of the aerosol representation methods was made in terms of the computational costs (Sect. 6.1), the mass concentrations of $O_3$, mineral dust and $PM_{2.5}$ (Sect. 6.2), the deposition of inorganics such as $SO_4^{2-}$, $NO_3^-$, and $NH_4^+$ (Sect. 6.3), AOT and light-absorbing AOT (AAOT) (Sect. 6.4), and CCN (Sect. 6.5). In Table 4, the statistical measures listed are the median simulation to observation ratio (*Sim:Obs*), correlation coefficient (*R*), and fractions of simulated values within a factor of two of the observed values (*Fa2*) of the relevant variables

such as $O_3$ and its precursor gas $NO_x$, dust extinction coefficient (Ext_D) and $PM_{10}$ during the dust events of April 2006





(defined later in Sect. 6.2), $PM_{2.5}$, wet deposition amounts of $SO_4^{2-}$, $NO_3^-$, and $NH_4^+$, AOT, SSA, total (dust and spherical) extinction coefficient (Ext_T), and $NC_{0.3}$. The discrepancies between the simulations and the observations are discussed in Sect. 6 for each corresponding variable, and the corresponding figures are shown in Supplement 3.

Table 4: Comparative statistical analysis of observed (Obs.) and simulated (Sim.) variables

| Variable | $N^a$ | Obs. Med.[b] | Unit | *Sim:Obs*[c] | | | $R^d$ | | | *Fa2*[e] | | |
|---|---|---|---|---|---|---|---|---|---|---|---|---|
| | | | | 5-ctg | 3-ctg | blk | 5-ctg | 3-ctg | blk | 5-ctg | 3-ctg | blk |
| $O_3^f$ | 3496 | 42.7 | ppbv | 0.91 | 0.91 | 0.91 | 0.63 | 0.63 | 0.63 | 0.92 | 0.92 | 0.92 |
| $NO_x^f$ | 3334 | 1.3 | ppbv | 0.94 | 0.94 | 0.92 | 0.40 | 0.39 | 0.39 | 0.48 | 0.48 | 0.48 |
| Ext_D$^g$ | 69 | 0.47 | $km^{-1}$ | 0.47 | n.a. | 0.42 | -0.17 | n.a. | -0.17 | 0.61 | n.a. | 0.42 |
| $PM_{10}^h$ | 69 | 370 | $\mu g\ m^{-3}$ | 0.77 | 0.77 | 0.74 | 0.22 | 0.23 | 0.25 | 0.58 | 0.58 | 0.58 |
| $PM_{2.5}^f$ | 695 | 8.83 | $\mu g\ m^{-3}$ | 1.08 | 1.22 | 1.15 | 0.57 | 0.57 | 0.62 | 0.82 | 0.81 | 0.76 |
| $PM_{2.5}^{f,i}$ | 695 | 8.83 | $\mu g\ m^{-3}$ | 0.81 | 0.78 | 0.92 | 0.58 | 0.57 | 0.62 | 0.68 | 0.68 | 0.74 |
| Precip.$^j$ | 253 | 60. | mm | 0.88 | 0.88 | 0.88 | 0.69 | 0.69 | 0.69 | 0.61 | 0.61 | 0.61 |
| W-nss-$SO_4^{2-\ j}$ | 253 | 0.77 | $mmol\ m^{-2}$ | 0.90 | 0.93 | 0.87 | 0.49 | 0.49 | 0.47 | 0.54 | 0.53 | 0.51 |
| W-$NO_3^{-\ j}$ | 253 | 0.80 | $mmol\ m^{-2}$ | 0.86 | 0.86 | 0.82 | 0.73 | 0.72 | 0.67 | 0.62 | 0.61 | 0.60 |
| W-$NH_4^{+\ j}$ | 253 | 0.67 | $mmol\ m^{-2}$ | 0.90 | 0.91 | 0.96 | 0.68 | 0.68 | 0.61 | 0.56 | 0.57 | 0.53 |
| AOT$^k$ | 566 | 0.3 | - | 0.42 | 0.46 | 0.42 | 0.39 | 0.40 | 0.38 | 0.44 | 0.47 | 0.46 |
| SSA$^k$ | 355 | 0.96 | - | 0.95 | 0.91 | 0.90 | 0.20 | -0.07 | -0.08 | 1.00 | 1.00 | 1.00 |
| Ext_T$^l$ | 4146 | 0.14 | $km^{-1}$ | 0.57 | 0.57 | 0.57 | 0.31 | 0.32 | 0.28 | 0.55 | 0.55 | 0.52 |
| $NC_{0.3}^m$ | 1630 | 9.91 | $\times 10^4\ L^{-1}$ | 1.17 | 1.15 | 1.50 | 0.36 | 0.33 | 0.29 | 0.64 | 0.63 | 0.55 |
| $PM_{10}^m$ | 1881 | 42.0 | $\mu g\ m^{-3}$ | 0.53 | 0.53 | 0.56 | 0.53 | 0.55 | 0.52 | 0.47 | 0.48 | 0.50 |

[a]Number of available data
[b]Median of observation data
[c]Simulation to observation median ratio
[d]Correlation coefficient
[e]Fraction of simulated values within a factor of two of the observed values
[f]Daily mean surface concentration
[g]Hourly mean median values below 300 m of extinction coefficient during the dust events in April 2006.
[h]Hourly mean values during the dust events in April 2006.
[i]$PM_{2.5}$ but compared with simulated $PM_{2.5}$ (pile-up)
[j]Half-monthly cumulative precipitation or wet deposition amount
[k]Daily mean column amount
[l]Daily mean median values below 300 m of extinction coefficient.
[m]Hourly mean surface concentration in April and December 2006 when the OPC data are available at Hedo.



# 6 Model performances for various purposes

In this section, the intermodule comparisons between the bulk equilibrium, 3-category, and 5-category methods are presented in terms of their relevant purposes, i.e., operational forecast, air quality forecast, and climate-relevant variable forecasts (such as AOT, CCN, and IN) of aerosols, respectively.

It should be noted here that the capability of the bulk equilibrium method in terms of operational forecast was not assessed by conducting forecast simulations (e.g., 3-day forecast with a data assimilation of initial conditions) but assessed in the hindcast simulation by comparing the relevant variables, which are used for operational forecasts such as surface mass concentrations of $O_3$, mineral dust, and $PM_{2.5}$, predicted by the bulk equilibrium method against those of the reference method, the 5-category method.

## 6.1 Computational costs

       The computational costs of the three methods are listed in Table 5. $R_{CPUtime}$ and $R_{aero}$ were measured for the earlier half-year simulations (i.e., 27 December 2005 to 1 July 2006). The nonequilibrium aerosol microphysics module consumed approximately half of the total CPU time for the 5-category method, while it consumed only 15% of that of the bulk equilibrium module (and the thermodynamic equilibrium model ISORROPIA2 consumed a major proportion of this 15%).
The total CPU times of the 3-category and 5-category methods are 44% and 91% greater than that of the bulk equilibrium module, respectively. For these three options, the number of gaseous species is the same (58), and the numbers of their aerosol attributes are 20, 38, and 50, respectively.

Table 5: Number of transport variables (gas plus aerosol attributes) ($N_{trans}$) and ratios of the total CPU time to that of the bulk
chemistry ($R_{CPUtime}$), as well as ratios of CPU time of the aerosol process to that of the total processes[1] ($R_{aero}$).

|  | $N_{trans}$ | $R_{CPUtime}$ | $R_{aero}$ (%) |
|---|---|---|---|
| Bulk equilibrium method | 78 | 1 | 15.0% |
| 3-category nonequilibrium method | 96 | 1.44 | 46.0% |
| 5-category nonequilibrium method | 108 | 1.91 | 54.9% |

[1] Excluding the I/O and MPI communications.



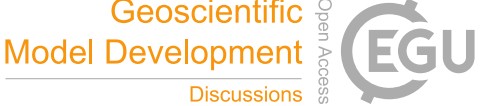

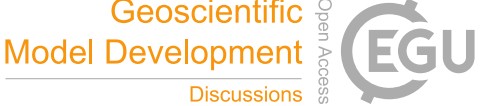

Figure 3: Seasonal mean surface concentrations of O₃ in (top to bottom) spring, summer, autumn, and winter of 2006 for (left to right) the bulk method, the 5-category method, and the ratios of the bulk to 5-category methods with surface wind vectors.

## 6.2 Surface mass concentration of air pollutants

Figures 3-5 present the seasonal mean surface mass concentrations of O₃, mineral dust, and PM₂.₅, which negatively impact the health of the population and the environment and thus are often used in operational air quality forecasts.

The mid-latitude westerlies are predominant in this region, and air pollutants are transported from west (or
10   northwest) to east (or southeast), associated with migrating disturbances, including the cold and warm frontal transports of





cyclones and anticyclonic transport throughout the year except during summer (e.g., Oshima et al., 2013). In summer, due to the Pacific high system, the transport of continental air masses is less dominant, and clean air masses of maritime origin are often transported to Japan. Northerly to northwesterly winds prevail in winter due to the Siberian high system. These seasonal wind patterns can be clearly observed in Figs. 3-5.

The simulated seasonal mean surface $O_3$ concentration over China was highest in the summer but that in Japan was highest in the spring due to the long-range transport associated with traveling disturbances (Fig. 3, Fig. S3-1). Figure S3-1 compares the observed and simulated daily $O_3$ concentrations with the precursor gas $NO_x$ concentrations at the Japanese EANET stations, and the statistical scores are shown in Table 4. The simulated medians agreed well with those observed medians (*Sim:Obs* of $O_3$ and $NO_x$ are approximately 0.9), although the correlation coefficients were not very large

(approximately 0.6 and 0.4 for $O_3$ and $NO_x$, respectively). The low correlation coefficients of $O_3$ were due mainly to the locations of the observation sites: *R* at remote sites was 0.76-0.91, whereas *R* at rural inland sites was low (0.24-0.49) due to the complex topography that cannot be resolved by the crude resolution of the current simulation. Low correlations of $NO_x$ were obtained mainly because it is a primary species. The uncertainty of emission flux and unresolvable heterogeneity of emission sources near the sites degraded the model performance of the primary species results more than they did for the

secondary species such as $O_3$. The difference in the horizontal distributions between the two methods was very small (Fig. 3). The differences between the bulk $O_3$ and 5-category $O_3$ were due mainly to the differences in the aerosol surface areas (different heterogeneous loss rates of $NO_x$, as a precursor of $O_3$ formation). It is inferred from Table 4 (which provides more significant digits) that the relative magnitudes of *Sim:Obs* of $NO_x$ and $O_3$ of the three methods are consistent (*Sim:Obs* of $NO_x$ of the bulk, 3-category, and 5-category methods were 0.925, 0.939, 0.938, and those of $O_3$ were 0.9108, 0.9115 and

0.9111, respectively). Further investigation is needed for quantitative assessment, but the errors of the bulk method were smaller than those of the other methods considered by up to 5%. Because this difference is minor compared to the difference between the simulation and observation results (especially for *R*), improving the model processes and/or boundary conditions will certainly improve all of the simulation results, regardless of the aerosol category methods that are selected.



Figure 4: Same as Fig. 3 but for mineral dust for (left to right) the ratios of the bulk to 5-category methods, the ratios of the 3-category to 5-category methods, and the 5-category method. The model topography is depicted in grayscale on the background of each panel. Topography is overlain by color shades for concentrations above the lower limit of the scale (in this case, 20 μg m⁻³).

The simulated surface mineral dust concentration was highest in the spring in the emission source regions (Taklamakan and Gobi) and the downwind regions such as Korea, and Japan (Fig. 4). Figure S3-2 presents the observed and simulated hourly $PM_{10}$ and Ext_D at Oki/Matsue and Hedo in April 2006, when mineral dust transport events were observed. Hedo (CHAAMS) includes EANET and AD-Net, whereas Oki (Matsue) is the only EANET (AD-Net) site. The distance between Oki and Matsue is 50-60 km and there are two model grids between the nearest grids to the sites. Table 4 compares the observed and simulated $PM_{10}$ and Ext_D during the dust events in the month. The dust events are defined for the period





in April 2006, when Ext_D and $PM_{10}$ are greater than 0.1 km$^{-1}$ and 300 μg m$^{-3}$ at the Oki/Matsue stations and 0.06 km$^{-1}$ and 150 μg m$^{-3}$ at the Hedo station. There are no data for Ext_D of the 3-category method because sea salt and dust particles are completely internally mixed so that the extinction coefficient of dust particles cannot be isolated from the extinction coefficient of COR, which includes abundant sea salt particles at these near-the-coast sites. The differences between the bulk and 5-category methods were generally lower than 5% and at most 10% (Fig. 4). These differences could be due mainly to the differences in the chemical aging of the mineral dust, which increases in hygroscopicity due to condensational growth during its long-range transport in the 5-category method. There are even more significant differences found between the 3-category and 5-category methods. The 3-category and 5-category methods both consider chemical aging processes, but the 3-category method assumes a complete internal mixture of dust and sea salt. Accordingly, the dust particles were more hygroscopic but smaller in the 3-category method compared to those in the 5-category method case; thus, with the 3-category method, the surface concentration was higher over the ocean in the spring by up to 10% due to the lower gravitational and dry deposition velocities. The 3-category to 5-category dust concentration ratio was higher in the regions on the continent close to the west lateral boundary due to the difference in the dust sizes; additionally, a complete dust/sea salt mixture was assumed at the boundary in the 3-category method. Because this difference (up to 10%) is minor compared to the difference between the simulation and observation (approximately 0.4-0.6 of *Fa2*, see Table 4), improving the model processes and/or boundary conditions will certainly improve all of the simulation results regardless of the aerosol category methods that are selected.







Figure 5: Same as Fig. 4 but for PM$_{2.5}$.

The differences in the model performance of the PM$_{2.5}$ prediction are shown in Fig. 5. This PM$_{2.5}$ was derived by Eq.
1, but the simulated PM$_{2.5}$ was sometimes defined as the pile-up of the mass concentrations of the dry components existing in the submicron categories, which is independent from the aerosol size distribution. The latter is denoted as PM$_{2.5}$ (pile-up) in the manuscript, and the horizontal distributions are shown in Fig. S4-1 in Supplement 4. The major difference between the simulated PM$_{2.5}$ and PM$_{2.5}$ (pile-up) is that PM$_{2.5}$ includes some proportion of natural aerosols, such as sea salt and dust, and can exclude proportions of submicron categories (i.e., ACM and AGR) larger than 2.5 μm, whereas PM$_{2.5}$ (pile-up) completely excludes sea salt and dust particles and includes all of the submicron categories (i.e., ATK, ACM and AGR in the 5-category method, ATK and ACM in the 3-category method, and SUB in the bulk method). The comparison between the





simulation and observation is shown in Fig. S3-3 and listed in Table 4. The performances of simulated $PM_{2.5}$ were comparable or better than those of $PM_{2.5}$ (pile-up), indicating consistent prediction of the aerosol size distributions. The $R$ values of $PM_{2.5}$ and $PM_{2.5}$ (pile-up) were similar. The *Sim:Obs* of the 3-category $PM_{2.5}$ (1.22) was larger than those of the other two methods due to the complete internal mixing assumption of the sea salt and dust (both mineral and anthropogenic)

particles in the 3-category method: more dust mass was included in $PM_{2.5}$ because the sea salt/dust mixture resulted in larger sea salt particles but smaller dust particles. The simulated $PM_{2.5}$ concentration over China was highest in the winter due to its higher emissions and more stable stratification, whereas in Japan, it was highest in the spring (Fig. 5). The 3-category $PM_{2.5}$ was generally lower than the bulk $PM_{2.5}$ but generally higher than the 5-category $PM_{2.5}$. Both the bulk $PM_{2.5}$ and 3-category $PM_{2.5}$ were larger than the 5-category $PM_{2.5}$, but for different reasons. The bulk $PM_{2.5}$ was higher due to larger submicron

$NO_3^-$, whereas the 3-category $PM_{2.5}$ was higher due to the assumption of a complete internal dust/sea salt mixture, as mentioned above. The bulk submicron $NO_3^-$ is larger because $NO_3^-$ can mix with either submicron particles (such as ATK, ACM, or AGR) or coarse mode particles (COR or DU) over the land in the 3-category and 5-category methods, whereas $NO_3^-$ is mixed with only SUB in the bulk method in the absence of sea salt particles. As found in Fig. S3-3, the large bulk to 5-category ratio of $PM_{2.5}$ (pile-up) was consistent with that of $PM_{2.5}$ for the same reason (i.e., larger submicron $NO_3^-$). In

contrast, the 3-category to 5-category ratio of $PM_{2.5}$ (pile-up) was negligibly small, indicating that the internal dust/sea salt mixture assumption has a minor impact on the dry mass concentration of the submicron categories. Table 4 shows that the *Sim:Obs* of $PM_{2.5}$ was greater than that of $PM_{2.5}$ (pile-up), indicating that approximately 20% of the sea salt (and 40% of the dust/sea salt mixture) contributed to the $PM_{2.5}$ at Oki and Rishiri. It is unclear if the bulk method is eligible for operational forecasting because the bulk $PM_{2.5}$ result was generally larger than the 3- and 5-category $PM_{2.5}$ results by approximately 20-

100% due to the abovementioned $NO_3^-$ mixture assumptions. Although they have not yet been discussed, there are still large discrepancies in the simulated and observed $PM_{2.5}$ chemical compositions in Japan. Because the uncertainties in both the simulated chemical compositions and their size distributions contribute to the simulated $PM_{2.5}$, it is not necessary for the 5-category method to yield the best prediction of $PM_{2.5}$. In fact, the $R$ value of the bulk $PM_{2.5}$ (0.62) was slightly higher than those of both the $PM_{2.5}$ and $PM_{2.5}$ (pile-up) of the 3- and 5-category methods (0.57-0.58) (Table 4). However, the current

analysis is insufficient to evaluate the eligibility of the model for $PM_{2.5}$ operational forecasting because the simulated $PM_{2.5}$ was compared with observations obtained at only two stations on remote islands. Currently, relevant work is ongoing under the framework of a multimodel intercomparison study in Japan called J-STREAM (Chatani et al., 2018).







Figure 6: Same as Figs. 4 and 5 but for the dry deposition of nss-SO$_4^{2-}$ and T-NH$_4^+$ plus T-NO$_3^-$.

## 6.3 Dry, wet, and fog deposition of acidic substances

Figs. 6 and 7 compare the predictions of the dry and wet depositions of the inorganic components of non-sea salt SO$_4^{2-}$ (nss-SO$_4^{2-}$, defined as [SO$_4^{2-}$] - 0.251 × [Na$^+$] (μg/m$^2$ or μg/m$^3$)), total (gas plus aerosol) NH$_4^+$ (T-NH$_4^+$), and T-NO$_3^-$, which are the major acidic and basic substances, obtained using the three methods. Once deposited on the ground surface, NH$_4^+$ is efficiently converted to NO$_3^-$ in the soil; thus, NH$_4^+$ is also an important agent for environmental acidification. The horizontal distribution of dry deposition was generally similar to that of surface concentration because the dry deposition flux is proportional to the surface concentration.





As shown in Table 4, the model precipitation predictions by NHM and the wet deposition amounts predicted by Chem are similar: 10-15% underestimation, approximately 0.7 of $R$, and 0.6 of *Fa2*, except 0.5 of $R$ for wet deposition of nss-$SO_4^{2-}$). A similar feature, underestimation of the wet deposition amounts, is found in Fig. S3-4. Certainly, this is not likely the only reason for the lower $R$ for nss-$SO_4^{2-}$, which could also be due to the effect of short-lived large sea salt

particles, as discussed in Kajino et al. (2012b). Because the Japanese EANET stations are located near the coasts (150-700 m from the coastline), the observed amount of W-$Na^+$ could be substantially affected by the short-lived (traveling distance is approximately 1 km) large sea salt particles ($D > 10$ μm), which is not considered in the model. As a result, the simulated W-$Na^+$ substantially underestimated the observation (*Sim:Obs* = 0.26-0.27). This discrepancy in $Na^+$ slightly lowered the $R$ of nss-$SO_4^{2-}$ compared to those of the other chemical components and precipitation (*Sim:Obs* of total (non-sea salt plus sea

salt)-$SO_4^{2-}$ = 0.42-0.44). $SO_4^{2-}$ from dimethyl sulfide (DMS), which was not considered in the simulation, could be a reason for the discrepancy, but the contribution may not be large for the region of the simulation (e.g., Kajino et al., 2004).

The dry depositions (Fig. 6) of the 3-category and 5-category methods were very similar (the difference was lower than 5% in most of the domain) because the size distribution and gas-aerosol partitioning of the components were similar. The separation of soot/soot-free and dust/sea salt particles in the 5-category method did not substantially alter the size or gas-

aerosol partitioning. The difference between the bulk and 5-category methods was significant and varied by up to a factor of two. This difference is mainly due to the gas-aerosol partitioning of T-$NO_3^-$. The differences in nss-$SO_4^{2-}$ and $NH_4^+$ simulated using different methods were less significant. The dry deposition of T-$NO_3^-$ of the bulk method was smaller over land because no coarse mode $NO_3^-$ was formed over land, where its dry deposition velocity is much larger than that of submicron $NO_3^-$. Consequently, compared to the other two methods, the total (i.e., nss-$SO_4^{2-}$, T-$NH_4^+$, plus T-$NO_3^-$) dry

deposition flux of the bulk method over land was smaller. The $HNO_3$ gas concentration of the bulk method was generally larger than those of the 3- and 5-category methods because the bulk method contained fewer counterpart ions of $NO_3^-$ (i.e., mineral dust) than the other two methods. Therefore, the dry deposition amount of $HNO_3$ gas (and so the total amount) of the bulk method was larger than that of the other two methods, especially over the ocean, where the dry deposition velocity of $HNO_3$ gas was much faster than that of aerosols.





Figure 7: Same as Fig. 6 but for wet deposition.

The horizontal distribution of the wet deposition flux is patchier than that of the dry deposition flux, but similar patterns were found for both fluxes (Fig. 7). The difference in the fluxes between the 3-category and 5-category methods was small (lower than 10-20% in most of the domain) but larger than the difference in the dry deposition fluxes. The wet deposition of the 3-category method was generally larger than that of the 5-category method due to the overestimation of wet deposition of $SO_4^{2-}$, $NH_4^+$, and $NO_3^-$ mixed with BC, as discussed later in Sect. 6.4.



Figure 8: Same as Figs. 6 and 7 but for fog deposition for (left to right) the bulk, the 3-category, and the 5-category methods.

Fog deposition could be significantly underestimated with a crude grid resolution model, such as in the current

5   study ($\Delta x$ ~10 km), although it has been significant in local-scale simulations ($\Delta x$ ~ 1 km) (e.g., Katata et al., 2011; Kajino et

al., 2019b). As the grid resolution becomes cruder, the elevation and slope of the model terrain are more flattened so that the

upslope fog formation is underestimated, and in many cases, the low-level clouds cannot reach the model ground surface. It

is well known that fog deposition over mountain forests is proportional to elevation (e.g., Katata et al., 2011). However,

implementation of a fog deposition scheme was a feature of the current model; the simulated inorganic deposition due to fog





is presented in Fig. 8. The fog deposition amounts were generally 1-2 orders of magnitude smaller than those of the dry and wet depositions. The fog deposition amounts of the bulk method are remarkably smaller than those of the other nonequilibrium methods. This is simply because submicron aerosols are smaller in the bulk method because the aerosol microphysical processes are not taken into account. Thus, a major proportion of the submicron aerosols of the bulk method

5  cannot be activated as fog droplets with such low supersaturation conditions for fog formation. The fog occurrence was higher in the colder seasons, and thus, the deposition areas were smallest in the summer, except in Taiwan. In Taiwan, fog deposition was highest in the summer. Larger deposition areas are found over the high topography with highly vegetated regions in the spring, autumn, and winter. The largest deposition areas are observed over the North China Plain in the winter, associated with the increase in the radiation fog frequency and air pollution levels in winter.

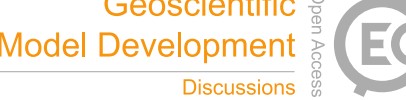

Figure 9: Same as Figs. 4-7 but for AOT (at 500 nm) below approximately 1 km AGL. The wind vectors are also averaged within 1 km AGL.

## 6.4 Aerosol optical thickness

Figures 9 and 10 compare the three methods in terms of their predictions of AOT and AAOT at a wavelength of 500 nm integrated from the ground surface up to approximately 1 km AGL, respectively.

The Mie theory calculation was performed to derive the AOT and AAOT at wavelengths of 500 nm by using simulated log-normal size distribution parameters and chemical compositions. The refractive indices of the Optical Properties of Aerosols and Clouds (OPAC) database (Hess et al., 1998) were used for the Mie calculation. For the refractive indices of the model components of UID, BC, OM, MD, NS, $SO_4^{2-}$, $NH_4^+$, $NO_3^-$, $Cl^-$, and $H_2O$, the databases of "insoluble",





"soot", "insoluble", "mineral", "sea salt", "water-soluble", "water-soluble", "water-soluble", "sea salt", and "fog" values, respectively, were used, with some modifications for BC to $1.85-0.71i$ (Bond et al., 2006) and MD to $1.5-0.001i$ (Aoki et al., 2005) at 500 nm. To calculate the optical properties of non-light-absorbing and light-absorbing mixtures, the Maxwell-Garnett approximation was used. The BC and MD masses were regarded as light-absorbing components, and the others were regarded as non-light-absorbing (certainly, some proportions of OM are light-absorbing, i.e., brown carbon, but they are not considered in the current model). The volume-weighted refractive indices were used for the light-absorbing and non-light-absorbing components in each category.

The comparisons between the simulated and observed AOT, SSA, and Ext_T are listed in Table 4, and the simulated and observed daily variations are compared in Fig. S3-5. The simulated AOT was underestimated by 50-60% (Table 4). There may be two reasons for this underestimation: the underestimation of upper atmospheric aerosol concentrations or the overestimation of the ratios of mass concentration to extinction coefficient (MEF: the mass/extinction conversion factor, as defined in Sugimoto et al., 2011). Considering that the simulated mass ($PM_{2.5}$, $PM_{2.5}$ (pile-up), and $PM_{10}$) quantitatively agreed with the observed mass ($PM_{2.5}$) or slightly underestimated the observed mass by 20-30% ($PM_{2.5}$ (pile-up) and $PM_{10}$) and that the degrees of underestimations of AOT and Ext_T were similar (50-60% and 45-50%, respectively), the underestimation of Ext_T could be due to the overestimation of MEF. The simulated SSA was underestimated by 5-10%, with an observed median of 0.96 (Table 4). Therefore, the values in Fig. 9 (AOT within 1 km) could be underestimated by approximately 50%, whereas the values in Fig. 10 (AAOT within 1 km) could be overestimated by 50-100%. One of the main purposes of the development of the 5-category method was to more realistically simulate the physical and chemical properties of BC-containing particles. The statistical scores of SSA predicted by the 5-category method are better than those predicted by the other two methods, which could support consistent implementation of the 5-category method.

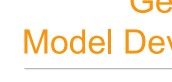
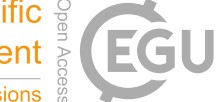

Figure 10: Same as Fig. 9 but for AAOT (at 500 nm) for (left to right) the 3-category method and the 5-category method and the corresponding ratio of the 3-category to 5-category methods.

5    Aerosol size distribution is an important parameter for aerosol-cloud-radiation interaction modeling, but the bulk method does not simulate aerosol microphysical processes. Therefore, the bulk method results are not discussed in detail, but the differences between the bulk method results and those from the other two methods considered are large (up to a factor of two, left panels of Fig. 9). It is only noted here that using different assumptions of size and mixing state (i.e., different settings among the three methods) could cause significant differences in the AOT simulation. This difference should be

10    taken into account when the operational forecast of mass concentration using the bulk method is associated with the data assimilations of optical properties such as AOT. In the large emission source regions and their downwind regions, the differences in AOT between the 3-category and 5-category methods were small, i.e., smaller than 10% in most of the





domain and seasons. The large difference over the Indian Ocean near the west and south boundaries of the domain in the summer was due to differences in the internal/external mixture assumptions of soot/soot-free and dust/sea salt at the lateral boundaries. In Fig. 10, the 3-category AAOT was slightly larger over the large emission source and their downwind regions (up to 10%). This was due to the overestimation of the so-called lensing effect (i.e., absorption

enhancement by a light-absorbing agent coated by a non-light-absorbing agent; e.g., Bond et al., 2006) by the 3-category method, as the soot particles are coated by all of the soot-free components. This is also true for the bulk method. Due to the larger submicron $NO_3^-$, the overestimation of the lensing effect should be most significant in the bulk method. This trend is also observed in Table 4: *Sim:Obs* of SSA of the 5-category, 3-category, and bulk methods were 0.95, 0.91, and 0.90, respectively. On the other hand, the internal mixture assumption of the 3-category method caused the

overestimation of the wet scavenging rates of soot and dust because the soot/soot-free and dust/sea salt particle mixture assumption could make the soot and dust particles larger and more hygroscopic. Consequently, the AAOT of the 3-category method was sometimes smaller than that of the 5-category method over the downwind regions, but this difference was small (up to 10%). Figure 11 shows the total (dry + wet) deposition of BC and the differences between the 3- and 5-category methods. Because the dry deposition velocity of BC is very small, most BC deposition occurred due to

wet deposition processes (also indicated by its patchy distribution). This difference was smaller in the spring and summer (reaching overestimation values of up to 10%) and largest in the winter (20-100%) due to the internal mixture assumption of soot and soot-free particles. Overestimation of the 3-category AAOT was more pronounced in the spring and summer than it was in the winter (Fig. 10) because condensation of secondary pollutants to BC was larger in the warmer seasons and because overestimation of the 3-category wet deposition of BC was more significant in the winter (Fig. 11). The

differences in the total deposition of mineral dust between the 3- and 5-category methods were very small.



Figure 11: Same as Figs. 6 and 7 but for the total (dry + wet) deposition of BC for (left to right) the 3-category method, the 5-category method, and the ratios of the 3-category to 5-category methods.





Figure 12: Same as Fig. 10 but for the averaged CCN number concentration below approximately 1 km AGL at a supersaturation of 0.1%.

## 6.5 Aerosol-cloud interactions

The horizontal distributions of the averaged CCN at a supersaturation value of 0.1% below approximately 1 km AGL are compared in Fig. 12. There are no CCN observation data available for the year, but the $NC_{0.3}$ observed by OPC was used alternatively for the model evaluations of simulated CCN. The critical diameter of 0.3 μm corresponds to 0.035-0.041% of the critical supersaturations for $\kappa$ of 0.3-0.4 (Petters and Kreidenweis, 2007). The OPC data are only available from March to April and November to December 2006 at Hedo. The statistical scores for $NC_{0.3}$ are listed in Table 4. Those of $PM_{10}$ for the



same period are also shown in Table 4 to compare the model performance for the prediction of number concentration against that of mass concentrations. The time series data are presented in Fig. S3-6. The simulated $NC_{0.3}$ overestimated the observation by 20-50%, especially for the peak concentrations (Fig. S3-6). The 3-category and 5-category methods successfully avoided the significant overestimations of the peak concentrations predicted by the bulk method, which

indicates consistent implementations of aerosol dynamic processes of the 3-category and 5-category methods. $PM_{10}$ at Hedo could be substantially affected by sea salt particles, whereas $NC_{0.3}$ could be controlled by the number of submicron particles. Thus, it is hard to evaluate the number-mass consistency with the available data, but we can safely conclude that the prediction of number concentration was more difficult than that of mass concentration, as the $R$ results of 0.29-0.36 for number concentration was smaller than those for mass concentration (0.52-0.55) (Table 4).

Whereas the differences in the total CCN between the two methods were very small (up to 10%) (Fig. 12), significant differences were observed for some components and occasions of CCN. Figs. 13-16 illustrate the scatter diagrams of the simulated daily mean, below-1-km-AGL-averaged, CCN (of the 3- and 5-category methods and at supersaturations of 0.1% and 1%) and mass concentrations of all of the model grids, excluding 10 grids from the four lateral boundaries. The numbers of points in the scatter diagrams were randomly reduced to one-twentieth of their value to avoid too many overlaps

of symbols. The two supersaturation levels of 0.1% and 1% were used for this evaluation as the typical ranges of stratiform and convective clouds (Seinfeld and Pandis, 2006).



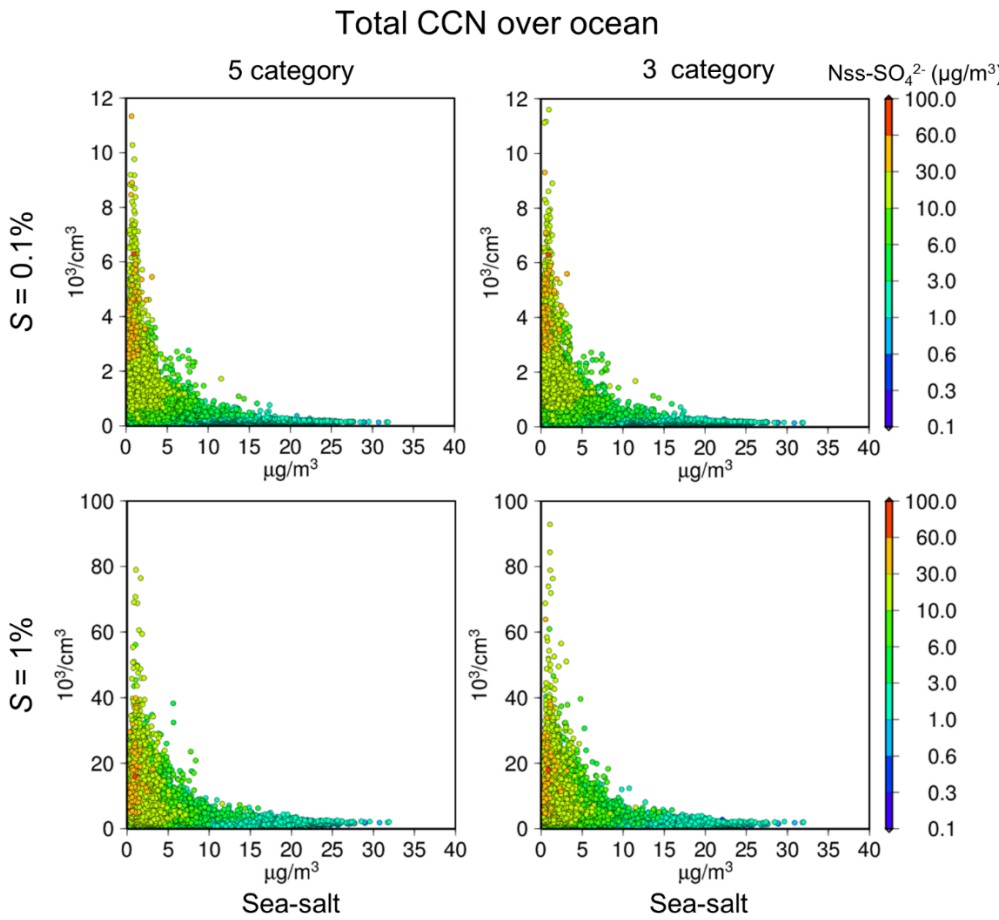

Figure 13: Scatter diagrams between the daily mean, averaged below 1 km AGL, total CCN number concentrations (y-axis) and dry mass concentrations of sea salt (x-axis) with nss-SO$_4^{2-}$ concentrations (colors) over the ocean grids for 2006. The number of data points was randomly reduced to approximately one-twentieth to avoid too much overlapping of the points at supersaturation conditions of (top) 0.1% and (bottom) 1% for (left) the 5-category and (right) 3-category methods.

Figure 13 shows the relationship between the total CCN and the sea salt mass over the ocean. In meteorological models, CCN spectra (i.e., the relationship between supersaturation and CCN) are often prescribed, and the CCN over the ocean is often assumed to be one order of magnitude lower than that over land (e.g., Rasmussen et al., 2002). However, as the ocean regions in the model domain were located downwind of the large emission source regions, the CCN could be as large as that over the land (e.g., Koike et al., 2012). The simulated total CCN over the ocean areas was up to two and three orders of magnitude larger than the simulated sea salt particles for both 0.1% and 1% supersaturations, respectively (the simulated total CCN reached up to approximately $10^4$ and $10^5$ cm$^{-3}$ at 0.1 and 1%, respectively, while number concentrations of sea salt particles were approximately 100 cm$^{-3}$ for the mass of 100 µg m$^{-3}$). The CCN at 1% was one order of magnitude larger than that at 0.1%. The CCN over the ocean was not correlated with the sea salt mass concentration. The CCN was proportional to the nss-SO$_4^{2-}$ concentrations, an indicator of anthropogenic secondary particles. These results indicate the





importance of considering spatiotemporal variations of CCN spectra in the meteorological models, rather than using the prescribed ones that depend only on the LUC (e.g., ocean or land). The CCN difference between the 3-category and 5-category methods was small, such that the difference in the aerosol-cloud interaction due to CCN changes predicted by the two methods over the ocean could also be small.

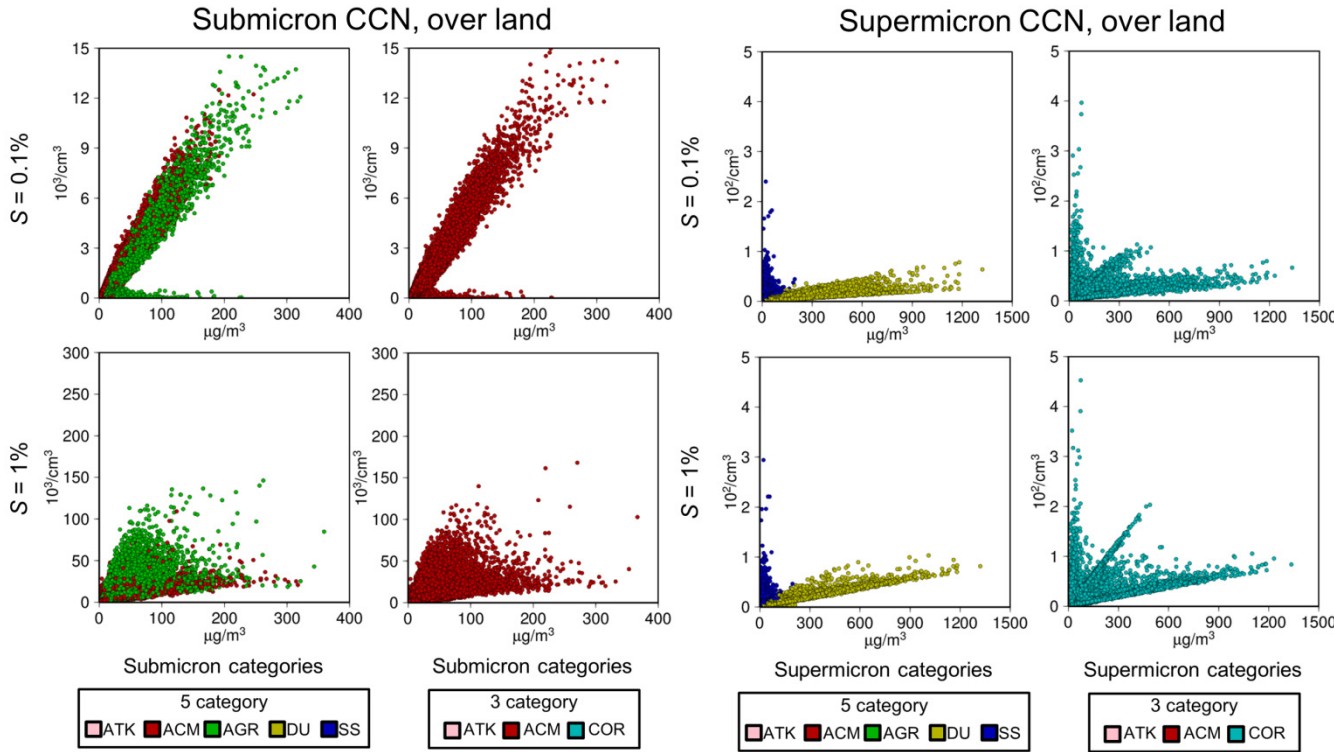

Figure 14: Same as Fig. 13 but for CCN over the land grids from (left four panels) submicron categories (i.e., ATK, ACM, and AGR for the 5-category method and ATK and ACM for the 3-category method) and (right four panels) supermicron categories (i.e., DU and SS for the 5-category method and COR for the 3-category method). The x-axes are the total dry mass concentrations of the submicron (left four panels) and supermicron (right four panels) categories. The colors of the dots correspond to those of the categories in the legends with the most dominant mass concentrations of the grids.

Figure 14 shows the differences in CCN over land. The submicron CCN at 1% was one order of magnitude larger than that at 0.1%. The difference in supermicron CCN between 0.1% and 1% was not very significant because the larger particles are CCN active at lower supersaturation. There was no difference for hygroscopic sea salt particles, whereas there was a slight difference for less hygroscopic dust particles. Due to the internal mixture assumption of soot and soot-free particles, the CCN at 0.1% of the 3-category method was slightly larger than that of the 5-category method. If the model assumes a constant size distribution of submicron particles, all of the points are aligned on a single line. Thus, the width of the aggregations of points indicates the uncertainty or variability of the simulated size distribution of aerosols. The CCN at



0.1% can vary by approximately 2-fold (e.g., it ranges from 3-6 $\times 10^3$ cm$^{-3}$ at 100 µg m$^{-3}$ of aerosol mass). The variation (or width) was much larger at 1%, reaching approximately 1 order of magnitude (5-80 $\times 10^3$ cm$^{-3}$ at 50 µg m$^{-3}$) because smaller particles can be activated and contribute less to the bulk mass concentration. The accurate prediction of the aerosol population is much more important for the submicron CCN at higher supersaturation values. For supermicron CCN, there are

5   apparent differences in the gradients of lines originated to mineral dust and sea salt in both methods. However, there is one intermediate line in addition to the dust and sea salt lines in the 3-category method, which is the result of the unrealistic internal mixture of dust and sea salt. Overall, it is safe to presume that the difference in the aerosol-cloud interaction due to CCN changes predicted by these two methods over land could be small due to the absence of sea salt particles.

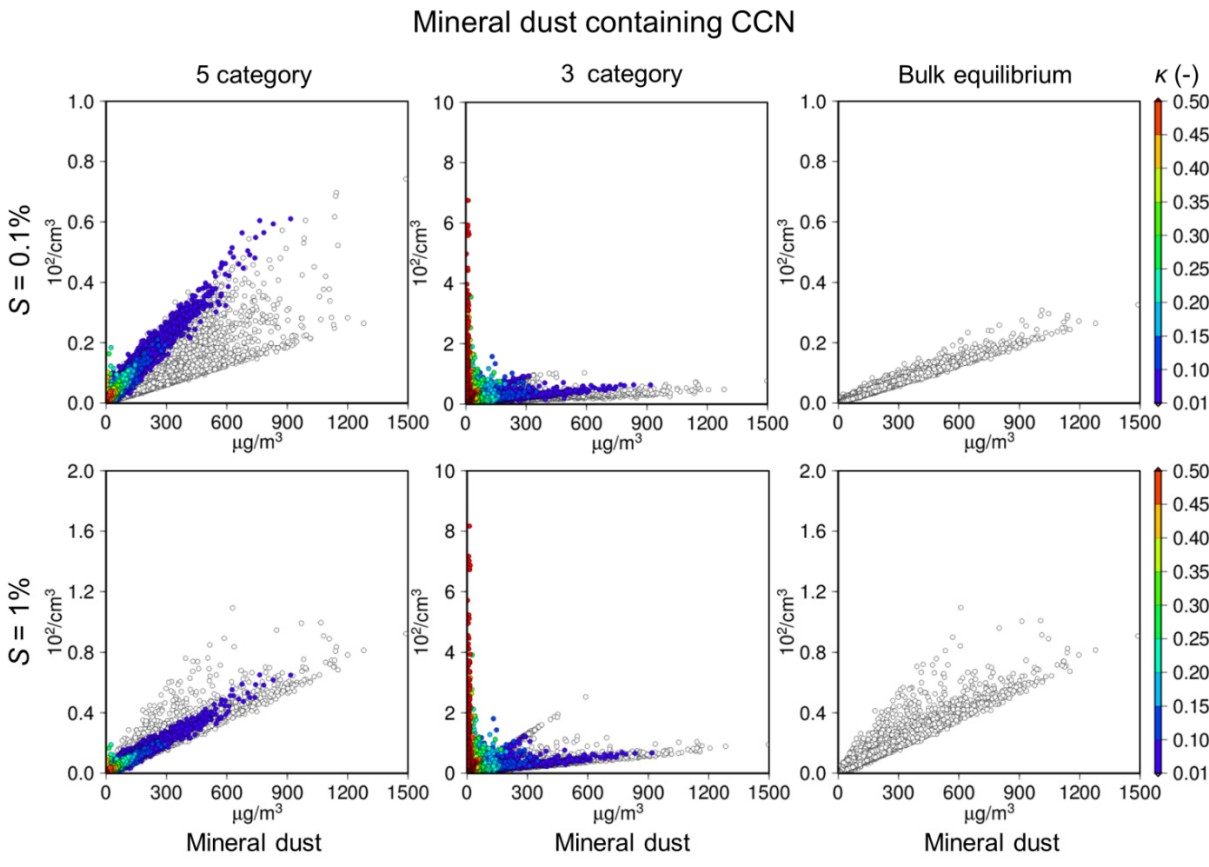

Figure 15: Same as Fig. 13 but for the mineral dust-containing CCN (y-axis) and mineral dust mass concentration (x-axis) with hygroscopicity $\kappa$ (colors) for (left to right) the 5-category, 3-category, and bulk methods. The circles are white if $\kappa$ is smaller than 0.01.

15       Figure 15 illustrates the relationship between mineral dust containing CCN and the mineral dust mass concentrations. Because dust is assumed to be inert and $\kappa$ is assumed to be zero in the bulk method, the CCN at 0.1% is proportional to only the mineral dust mass concentration, while the variation in the size distribution is more important for the





CCN at 1%. Because the hygroscopic growth of mineral dust is considered in the 5-category method, the dust-containing CCN is up to several times larger than that of the bulk method (from 0.1 to 0.4 ×10² cm⁻³ at 600 μg m⁻³). As previously mentioned, the unrealistic internal mixture of dust and sea salt in the 3-category method caused unrealistic gradients, i.e., one that is colored in red (sea salt with tiny dust mass) and a line in between (approximately at $x$ = 600 μg m⁻³ and $y$ = 2.5 ×10²

5    cm⁻³) that can be clearly seen at 1% (sea salt/dust mixture). The simulated difference in dust-containing CCN would have almost no impact on CCN-cloud-precipitation interaction processes because its number concentration was much smaller than that of submicron particles. As mineral dust is an efficient IN (e.g., Lohmann and Diehl, 2006), it should be noted here that this difference can have significant impacts on IN-cloud-precipitation interaction processes, especially with immersion and condensation freezing (i.e., the freezing of supercooled water droplets containing ice nuclei).

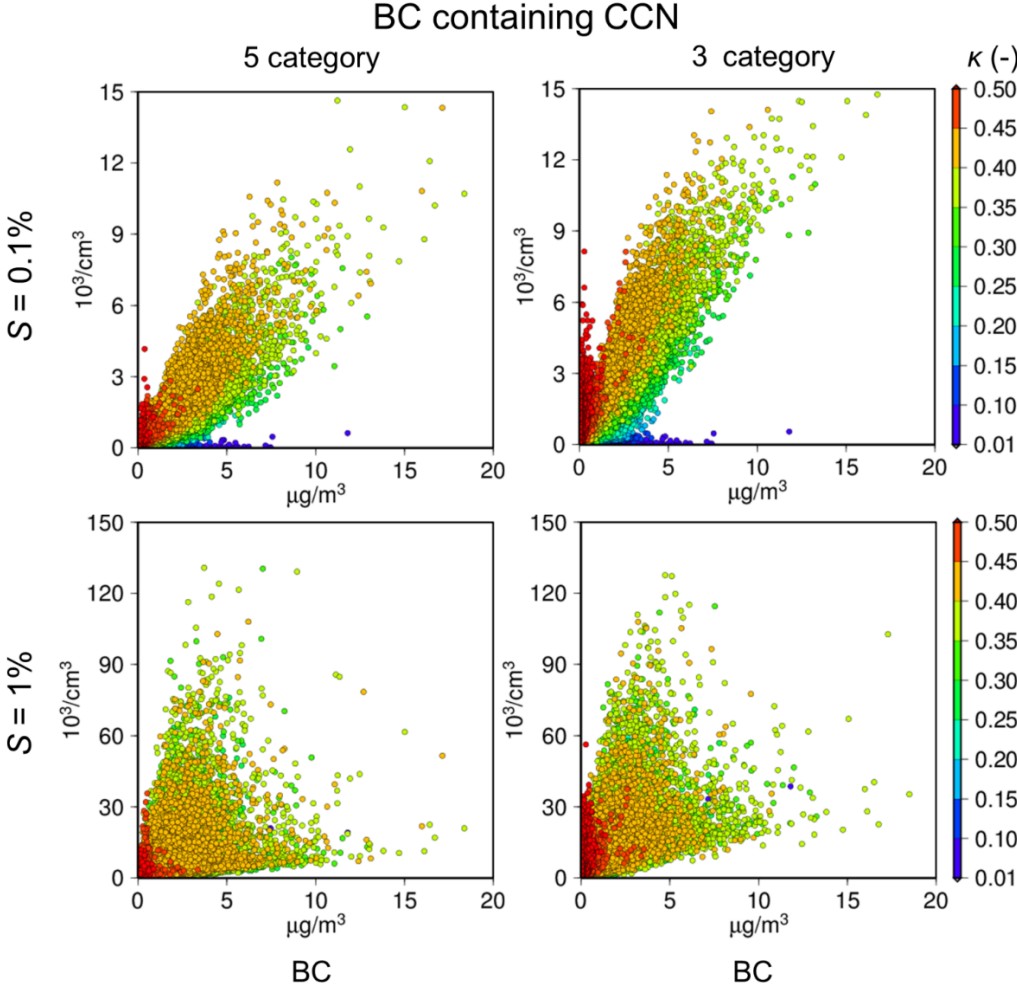

Figure 16: Same as Fig. 15 but for the BC-containing CCN (y-axis) and BC mass concentration (x-axis) with hygroscopicity $\kappa$ (colors) for (left) the 5-category and (right) 3-category methods.



Despite the external mixture treatment of soot and soot-free particles in the 5-category method, the difference in the total submicron CCN between the 5- and 3-category methods was found to be minor (Fig. 14). However, the simulated BC-containing CCN values of the two methods were different, as shown in Fig. 16. The BC-containing CCN of the 3-category method at 0.1% was generally 1.5 times larger than that of the 5-category method (e.g., up to 9 and 6 $\times 10^3$ cm$^{-3}$ at 4 μg m$^{-3}$,

respectively) because the absence of BC-free particles causes excess amounts of coatings on BC-containing particles (Oshima et al., 2009b). This difference caused higher BC deposition to be simulated by the 3-category method, as presented in Fig. 11. However, the difference in the BC-containing CCN of the two methods at 1% was not as large as that at 0.1% because the CCN number concentration at higher supersaturation was not affected by hygroscopicity.

## 7 Conclusion

This study provides a comparison of aerosol representation methods implemented in a regional-scale meteorology-chemistry model (NHM-Chem). Three methods are currently available: the 5-category nonequilibrium (Aitken, soot-free accumulation, soot-containing accumulation, sea salt, and dust), 3-category nonequilibrium (Aitken, accumulation, and coarse), and bulk equilibrium (submicron, sea salt, and dust) methods. The 3-category method is widely used in three-dimensional air quality models. The 5-category method, the standard method of NHM-Chem, was an extensional development of the 3-category

method to improve the predictions of regional climate, including aerosol-cloud-radiation interactions, by implementing separate treatments of light absorbers and ice nuclei, namely, soot and dust, from the accumulation and coarse mode categories. The bulk equilibrium method was developed for operational air quality forecasting with simple aerosol dynamics representations. The number of tracers in the 5-category, 3-category, and bulk methods are 108, 96, and 78, respectively (they each contain 58 common gases and 50, 38, and 20 aerosol attributes, respectively). The total CPU times of the 5-

category and 3-category methods were 91% and 44% greater than that of the bulk method, respectively.

The bulk equilibrium method was shown to be eligible for operational forecast purposes, namely, the surface mass concentrations of air pollutants such as O$_3$, mineral dust, and PM$_{2.5}$. The differences in the simulated seasonal mean concentrations between the bulk method and the reference method, i.e., the 5-category method, were smaller than 5% and 5-10% for O$_3$ and mineral dust, respectively. The initial and boundary conditions should be improved before model

formulation. The difference in PM$_{2.5}$ was large, i.e., up to 20-100%. Improving the model formulation, as well as its initial and boundary conditions, is needed.

The 3-category method was shown to be eligible for air quality simulations, namely, surface concentrations and depositions of bulk chemical species. The differences in the simulated seasonal mean bulk mass concentrations (PM$_{2.5}$ (pile-up)) and dry and wet depositions of major inorganic components (SO$_4^{2-}$ + NH$_4^+$ + NO$_3^-$) between the 3-category method and

the reference method were smaller than 5%, 5%, and 10%, respectively. However, the internal mixture assumption of soot/soot-free and dust/sea salt particles in the 3-category method resulted in significant differences in the size distribution





and hygroscopicity of the particles. The differences in PM$_{2.5}$, defined by the simulated size distribution, between the 3- and 5-category methods were 20-100%.

Although the 3-category method was not designed to simulate aerosol-cloud-radiation interaction processes, its performance in terms of total properties, such as AOT and CCN, was acceptable compared to that of the reference method (i.e., both differences were smaller than 10%). However, some specific parameters exhibited significant differences or systematic errors. For example, the unrealistic dust/sea salt complete mixture of the 3-category method induced significant errors in the prediction of mineral dust-containing CCN. The overestimation of soot hygroscopicity by the 3-category method induced errors in BC-containing CCN at a supersaturation of 0.1% (overestimation by ~50%), BC deposition (overestimation by 20-100%), and AAOT (difference of < 10%). In contrast, the difference in AAOT was less pronounced because the overestimation of the absorption enhancement was compensated by the overestimation of hygroscopic growth and the consequent loss due to in-cloud scavenging.

For the operational forecast, the simulation time is critical; thus, the bulk method is recommended. On the other hand, for research purposes, because the difference in the computational resources between the 5-category and 3-category methods is not very large (44% in CPU time and 12.5% in number of tracers, i.e., memory and storage), the 5-category method is regarded as the standard (default) aerosol option of NHM-Chem.

The modal approach was used in the current aerosol microphysics module, but the same conclusion can also be inferred for the sectional (or bin) approaches in terms of the advantage of the 5-category approach, namely, the soot and dust sections should be separated from the light-scattering hygroscopic particle section for detailed investigations of aerosol-cloud-radiation interactions.

There is a 12-category method that was implemented in a regional model (Glassmeier et al., 2017). This is currently the ultimate representation of the category method in 3-D CTMs. It is necessary to compare our 3- and 5-category methods to the 12-category approach as a benchmark simulation. The optimization of computational efficiency against complexity (and accuracy) of the aerosol mixing state representations should be achieved via particle analysis studies performed by electron microscopy and particle-resolved models. There are still large discrepancies between the simulation and observation results. Furthermore, model evaluations should be made in the near future with respect to spatial distributions (using satellites), vertical profiles (using sonde, aircraft, satellite, and lidar), and aerosol size distributions (using such as scanning mobility particle sizer (SMPS), aerodynamic particle sizer (APS), polarized optical particle counter (POPC) methods) for improving NHM-Chem. In particular, model validations using the size distribution and vertical profile measurements are indispensable to identify the reason for the underestimation of AOT and Ext_T by the current model. There have been tremendous advancements in new particle formation, organic chemistry, and ice nucleation studies. This new knowledge should be timely and properly reflected in the modeling framework. The aerosol module comparison should be performed again in the future after substantial improvements in NHM-Chem are made, as the sensitivity could be different from the current simulations. The sensitivity of aerosol-cloud-radiation interactions to the three aerosol modules will be assessed after the installation of feedback processes to the online coupled NHM-Chem.



**Code and data availability**

The NHM-Chem source code is available subject to a license agreement with the Japan Meteorological Agency. Further information is available at https://www.mri-jma.go.jp/Dep/glb/nhmchem_model/application_en.html (last access: 1 October 2020). The simulation results are free to use. The simulated and observed data used in the manuscripts are available at

https://mri-2.mri-jma.go.jp/owncloud/s/ASBzHdtqy9ZpbB4 (last access: 14 September 2020). The raw observation data of EANET, SKYNET, and AD-Net can be obtained from the web sites as described in Sect. 5.

**Author contribution**

MK designed the current research. MK mainly developed the offline coupled model with MD, who developed its online coupled version. MK and MD collaborated with TTS, NO, KY, TYT, JC, AH, MI, AK, YI, SS, KA, YZ, YI, and TM during

model development. MK performed the simulations, which were evaluated by the observation data provided by TY, MM (Miyashita), PK, AS, and HI. YI, HU, TM, and MM (Mikami) cosupervised this study.

**Competing interests**

The authors declare that they have no conflict of interest.

**Acknowledgments**

This work was mainly supported by the Fundamental Technology Research of MRI (M5 and P5), the Integrated Research Program for Advancing Climate Models (TOUGOU) Grant Number JPMXD0717935561 and Arctic Challenge for Sustainability II (ArCS II), Grant Number JPMXD1420318865 from the Ministry of Education, Culture, Sports, Science, and Technology Japan (MEXT), and Research Institute for Humanity and Nature (RIHN: a constituent member of NIHU) Project No.14200133 (Aakash). This work was also supported by the Japanese Society for the Promotion of Sciences (JSPS)

KAKENHI Grant Numbers JP19H01155, JP16KK0018, JP19F19402, JP19H02649, JP18H03363, JP18H05292, and JP2000636) and the Environment Research and Technology Development Fund (JPMEERF20165005, JPMEERF20172003, JPMEERF14S11203, JPMEERF20202003, and JPMEERF20205001) of the Environmental Restoration and Conservation Agency of Japan and in part by a grant for the Global Environmental Research Coordination System from the Ministry of Environment, Japan (MOEJ). The authors thank Dr. Toshinori AOYAGI of the JMA, Prof. Mitsuo OHIZUMI of

Meteorological College, and Drs. Hiroaki NAOE, Masashi NIWANO, and Masaya NOSAKA of the MRI for their useful comments on model developments. The authors are deeply grateful for the support of Mr. Shohei SUZUKI and Mr. Kazuki ITO of the University of Tsukuba for the visualization of simulation and observation data and Ms. Kyoko KANEBA for the



comments on the manuscript. The authors also thank the Kagoshima Meteorological Office and Aso Volcano Disaster Prevention Council for providing the volcanic $SO_2$ emission measurement data of Mt. Sakurajima and Mt. Aso, respectively.

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
