# Peer review of "Comparison of three aerosol representations of NHM-Chem (v1.0) for the simulations of air quality and climate-relevant variables"

_Geoscientific Model Development, 2020_

## Referee Comment (RC1) · Anonymous Referee #1 · 2 Nov 2020

The paper by Kajino et al. 2020 primarily compares 3 existing aerosol representation schemes with varying complexity, namely a simple bulk, a 3- and a 5-category method within the newly developed chemical transport model (CTM) NHM-Chem. In a previous version of the paper, one of the key shortcomings was the missing link to the complete description of the model system, or the poor description within respectively. With the general model description paper now being published, the existing study gains in quality and also presents a relevant topic which itself fits to the scope of the journal. The language does not need significant review.

With the general functionality, the technical realizations and differences in aerosol rep-

resentations being elaborately discussed, an overall synthesis is missing which provides clear statements of the potential of the three themes discussed. Beginning with the heading, it does not come out clearly what is meant by 'air quality' and 'climate-relevant' variables and how that difference is tackled within the study. That aspect should be highlighted better in the introduction and within the discussion/conclusion. The overall quality of the paper has pretty much increased compared to previous versions. The points however which still need further work will be pointed out in the following:

Abstract:

If online coupling is not done within that study, it should be removed from the abstract. That aspect however is important when discussing the shortcomings and the outlook.

Introduction Page 2, Line 21: unclear: decrease air concentration

2, 26-31: re-write avoiding repetitions 'aerosols'

3, 17: are developed; the terms regional climate, air quality and operational forecasting should be explained more detailed, also highlighting how each single aspect has been addressed in the paper

3,20-25: unclear, whether the bulk and the 5-category schemes have been developed in the course of the study or have been existing before

NHM-Chem

4,11: better model configuration than schemes of the CTM

Aerosol representation

8,3: unclear: 'fully solve for'

8,10: unclear whether data assimilation is done here. That aspect is important when discussing the model's potential for operational forecast.

Interactive
comment

Setup 11,15: How where the two datasets combined, please specify

Model performance

In the beginning of the chapter, it has to be clarified which different aspects are considered in terms of the relevant purposes operational forecast, air quality forecast and climate forecast. How are these aspects discussed in that paper? What are the differences between studied processes, variables or even model configuration? The R-Values for PM10 are particularly low. Please discuss that aspect in terms of model performance for operational forecast, also highlighting the differences to the performance for PM2.5. It is partly discussed in the text, but more clarity is needed.

Figure 3: white areas in left and middle panel (also in Figure 12)?

20,8: show simulated medians in Table 4

20,11: specify 'remote sites'?

20,12: What are the key problems in the underestimation of NOx here? Problems with the emission dataset or chemical origin? Please further discuss that aspect with regard for using that model system in 'operational mode'. What is the ratio between NO/NO2 in total NOx?

24,20: discuss the large spread 20-100%

26,16: Why is that aspect particularly pronounced over sea areas? Figure 6: Why is 'Bulk' so much higher over the sea?

27,4: reason for patchiness?

Conclusion

As mentioned earlier, the conclusion is still missing a clear synthesis, which in places also results from missing details at various places in the manuscript. The authors are encouraged to address the following points:

- P 41, Line 21: How was the operational forecast quality assessed? It is unclear if the term 'operational forecast' simply relates to the selected variables or also includes a change in the model setup (how is DA addressed?)

- P 41, Line 24: How exactly should the initial and boundary conditions be improved?

- P 41, Line 25: Referring to your model results: where are the biggest shortcomings?

- P42, Line 24: See point above. Summarize dominant reasons for discrepancies.

- P42, Line 31: what is meant by timely and properly reflected? What are the future plans? Despite the shortcomings; what are the key benefit of the current configuration presented in this paper? What should be the core areas of future development?

---

## Referee Comment (RC2) · Anonymous Referee #2 · 19 Jan 2021

The paper of Kajino et al. compares the three aerosol representations (bulk, 3-category, 5-category) which are available in the regional CTM NHM-Chem. The results of air quality related variables (O3, NO2 etc) are compared with observations. Further, CCNs and AOT etc. are compared between the three aerosol representations. The paper is a re-submition of a paper which was rejected earlier in GMD.

I already reviewed the rejected earlier submission of the paper. The focus has now changed from a model description paper to an model evaluation paper. The manuscript fits well into the scope of the journal. The paper is very long and parts of it could be shortened, while some parts could be improved with adding some more discussion.

[Figure]

Further, to my opinion the language should be improved at some points. In general, the paper needs some larger revisions before it can be accepted for publication.

Major comments:

To my opinion the figures are partly confusing. Some of the figures show Bulk/5-cat, 3-cat/5-cat, 5-cat, other show 3-cat, 5-cat, 3-ca/5-cat. An example are Fig. 3 and Fig.4. To add a little bit of confusion the caption of Fig 4 says (same as Fig. 3). I think it would be much easier if all figures would have the same design. Similar for example Fig 8 and 6/7 or Fig 9 and Fig 10.

Currently, there is no coupling of meteorological variables and aerosol, which is a major shortcoming of the study. This should be clearly mentioned and discussed (see also Reviewer 1).

The authors included a lot of information to the supplement, but sometimes discuss this supplementary material very detailed. As an example see page 20 l5ff. Either this are supplementary information (what is fine) or this are no supplementary information. If this are no supplementary information the figures should be included into the manuscript. If this are supplementary information the long description should be moved to the supplement to shorten the paper a little bit.

Generally, the paper offers a lot of analyses and information, but the results are of course only valid for the NHM-CHEM model. Therefore, I ask the authors to shorten parts of the general description of the results a little bit and put more empathize on the following points:

1) why are these variables important (see reviewer 1) 2) Discuss similarities /differences of the aerosol representations available in NMHC-Chem with aerosol representations available in other models 3) Discuss which findings most important for other aerosol models

Minor comments:

[Figure]

P4l20ff: The discussion about what schemes/couplings have been used in which publications is where long and could be removed without loosing important information for the paper

The description in p12l5ff about the temporal length of the simulations is very confusing and should be rephrased

The sentence 'We applied the monthly mean values of GFED3 without temporal variations" is unclear(the same monthly means in each month?)

The part in the conclusion: ' The initial and boundary conditions should be improved before model formulation. The difference in PM2.5 was large, i.e., up to 20-100%. Improving the model formulation, as well as its initial 25 and boundary conditions, is needed.' is unclear. What exactly and why should initial and boundary information be improved?

I suggest to rename the conclusion into 'conclusion & Discussion'

---

## Author Comment (AC1) · 22 Feb 2021

Thank you very much for your constructive comments and your time for RC1. We have already revised our manuscript according to your comments and so here we attached the file containing the point-by-point responses to your comments, which will also be used for your second review. Thank you very much for your understanding.

Please also note the supplement to this comment:
https://gmd.copernicus.org/preprints/gmd-2020-229/gmd-2020-229-AC1-supplement.pdf

[Figure]

**Supplement:**

Dear anonymous referee #1,

We very much appreciate your constructive comments and your time for RC1. Thanks to your review, our manuscript was substantially improved, especially for clearness of the sentences. All your comments were taken into account in the revised manuscript.

Point-by-point responses to your comments are written in blue in this letter.

With best regards,
Mizuo Kajino
* * *
[1] The paper by Kajino et al. 2020 primarily compares 3 existing aerosol representation schemes with varying complexity, namely a simple bulk, a 3- and a 5-category method within the newly developed chemical transport model (CTM) NHM-Chem. In a previous version of the paper, one of the key shortcomings was the missing link to the complete description of the model system, or the poor description within respectively. With the general model description paper now being published, the existing study gains in quality and also presents a relevant topic which itself fits to the scope of the journal. The language does not need significant review.

With the general functionality, the technical realizations and differences in aerosol representations being elaborately discussed, an overall synthesis is missing which provides clear statements of the potential of the three themes discussed. Beginning with the heading, it does not come out clearly what is meant by 'air quality' and 'climate-relevant' variables and how that difference is tackled within the study. That aspect should be highlighted better in the introduction and within the discussion/conclusion. The overall quality of the paper has pretty much increased compared to previous versions. The points however which still need further work will be pointed out in the following:

[1] Thank you for the evaluation and comment. We clearly defined these terms in 1. Introduction and 7. Conclusion and discussion (the section name was changed according to RC2 [10]), respectively, as follows:

(Introduction) "In this study, surface concentrations and depositions are referred to as

air quality variables, whereas variables involved in aerosol feedback processes such as optical properties and cloud and ice nucleation properties are referred to as climate-relevant variables."

(Conclusion and discussion) "The three methods were intercompared for the predictions of air quality and climate-relevant variables. In this study, surface concentrations and depositions are referred to as air quality variables, whereas variables involved in aerosol feedback processes such as optical properties and cloud and ice nucleation properties are referred to as climate-relevant variables."

*Abstract:*

[2] If online coupling is not done within that study, it should be removed from the abstract. That aspect however is important when discussing the shortcomings and the outlook.

[2] Thank you for your comments. We removed the relevant sentence from the abstract and moved it to the end of the "Conclusion and discussion" section. We also inserted the following sentence in the 2nd paragraph of "Conclusion and discussion" section: "(implementation of aerosol feedback processes to NHM-Chem is still ongoing)."

*Introduction:*

[3] Page 2, Line 21: unclear: decrease air concentration

[3] (1st paragraph of Introduction) Thank you for pointing this out. The sentence was awkward. If the removal rates of aerosol increase, deposition increases and air concentration decreases. We rephrased the relevant sentences as follows:

"The removal rates of aerosols, which alter atmospheric life time and earth surface contaminations, depend highly on these properties"

[4] 2, 26-31: re-write avoiding repetitions 'aerosols'

[4] (2nd paragraph of Introduction) We avoided repetitions by rephrasing from "properties of aerosols" to "their properties". If one sentence includes two "aerosols", the latter was changed to "those".

[5] 3, 17: are developed; the terms regional climate, air quality and operational forecasting should be explained more detailed, also highlighting how each single aspect has been addressed in the paper

[5] (last paragraph of Introduction) We changed from "developed" to "are developed". We totally removed the unclear phrase "regional climate" throughout the manuscript and changed it to more concrete expressions such as "aerosol-cloud-radiation processes" or "aerosol feedback process". The relevant sentences are reorganized as follows:

"The three aerosol representations are developed for the three respective purposes of predictions, aerosol-cloud-radiation interaction processes (or aerosol feedback processes), air quality issues (surface air concentrations of hazardous materials including their depositions), and operational forecasting (real-time forecast of hazardous materials concentrations with high computational efficiency)"

[6] 3,20-25: unclear, whether the bulk and the 5-category schemes have been developed in the course of the study or have been existing before

[6] (last paragraph of Introduction) It has been existing before, since Kajino et al., J. Meteor. Sci. Japan (2019). We modified the relevant sentence from "From the context mentioned above, … three options for aerosol representations … *are implemented in a model and intercompared* in this study" to "From the context mentioned above, … three options for aerosol representation … *already implemented in a model are intercompared* in this study".

*NHM-Chem:*

[7] 4,11: better model configuration than schemes of the CTM

[7] (3rd paragraph of Sect. 2) We changed it.

*Aerosol representation:*

[8] 8,3: unclear: 'fully solve for'

[8] (5th paragraph of Sect. 3) "Fully" meant nucleation, condensation, coagulation, and deposition, but we realized that it is not a general term. We think that "fully" was not needed here and so we modified the sentence as follows: "The 5-category and 3-category methods solve for aerosol microphysical processes by using the …".

[9] 8,10: unclear whether data assimilation is done here. That aspect is important when discussing the model's potential for operational forecast.

[9] (5th paragraph of Sect. 3) Sorry for the confusion. Data assimilation is applied for the

operational forecast, but it was not applied for the results presented in the paper because the purpose of the paper is the comparison of aerosol representations. We inserted the following sentence to the relevant paragraph: "It should be noted here that the data assimilation was not applied to the simulations, because the current study focused on variations in the model performances due to the different aerosol representations. The same initial and boundary conditions were used for the all simulations."

*Setup:*

[10] 11,15: How were the two datasets combined, please specify

[10] (1st paragraph of Sect. 4.1) It was described later in the same paragraph. To avoid this confusion the sentence was rephrased to "two half-year CTM simulations were conducted and then combined due to the initialization issue regarding the land surface model". Please refer to our reply to RC2 [7], which substantially enhanced the clarity of the data handling.

*Model performance:*

[11] In the beginning of the chapter, it has to be clarified which different aspects are considered in terms of the relevant purposes operational forecast, air quality forecast and climate forecast. How are these aspects discussed in that paper? What are the differences between studied processes, variables or even model configuration?

[11] (1st paragraph of Sect. 6) Thank you for your valuable comment. In the first paragraph, the terms "operational forecast", "air quality forecast", and "climate forecast" are clearly defined with improvements of terms as follows:

"in terms of their relevant purposes, i.e., simulations of variables often used for operational forecast (such as $O_3$, mineral dust, and $PM_{2.5}$), simulations of air quality variables (surface concentrations and depositions of pollutants), and simulations of climate-relevant variables (such as AOT, CCN, and ice nucleating particles (INP)), respectively".

We also added the new paragraph in the end of Sect. 6, which clearly described the objectives of the section as follows:

"The main objectives of this section are itemized as follows: (1) to compare the computational efficiency of the three methods, (2) to quantify the deviations of the

widely used 3-category method and the efficient bulk equilibrium method from the most realistic aerosol representation of NHM-Chem, the 5-category method, and (3) to assess the discrepancy between the simulated and observed variables and how the discrepancy varied depending on the three methods" The similar statement was repeated in the 2nd paragraph of Sect. 7 "Conclusion and discussion".

[12] The R- Values for PM10 are particularly low. Please discuss that aspect in terms of model performance for operational forecast, also highlighting the differences to the performance for PM2.5. It is partly discussed in the text, but more clarity is needed.

[12] $R$ values for $PM_{10}$ are particularly low, because those are the comparisons of hourly concentrations during the dust events at two stations (totally 69 data). Deviations in durations of simulated and observed dust events caused significant low correlations among them. In contrast, $R$ for daily $PM_{10}$ for the whole year at all stations were approximately 0.6, as presented in Table 4 of Kajino et al., JMSJ (2019), which was comparable with $R$ for daily $PM_{2.5}$. It is clearly stated in the 4th paragraph of Sect. 6.2 and to avoid confusion the variable name of $PM_{10}$ is changed to $PM_{10}$_D in the revised manuscript. Time resolution is also added in Table 4. $PM_{10}$_D is defined in Sect. 5 when it is appeared first time.

The following sentences are inserted in the 4th paragraph of Sect. 6.2 of the revised manuscript:

"Table 4 compares the observed and simulated Ext_D and PM10_D during the dust events in the month. The R values for Ext_D and PM10_D are particularly low, mainly because the values are the comparisons of hourly concentrations during the limited period (totally 69 data). The R values for the daily concentrations of Ext_D and PM10 at all stations for the whole period are available in Table 4 of Kajino et al. (2019a). The R value for Ext_D was still low (0.25) but that for $PM_{10}$ were 0.57-0.58, comparable with other variables such as $PM_{2.5}$ and $O_3$."

Please note that this part was moved to 3rd paragraph of Supplement 3, according to RC2 [5].

[13] Figure 3: white areas in left and middle panel (also in Figure 12)?
[13] According to RC2 [3], Fig. 3 was modified so that the topography was depicted under shades. There are no white areas in the new figures. There were no white areas in Fig. 12 (Fig. 11 in the revised manuscript). Areas below the lowest value (10 $10^1$/cm$^3$

for the Fig. 11's case), which appeared white indicating topography height.

[14] 20,8: show simulated medians in Table 4
[14] We added the simulated medians in the table.

[15] 20,11: specify 'remote sites'?
[15] (3rd paragraph of Sect. 6.2) We added the stations names, "Rishiri, Sado, Oki, Ogasawara, and Hedo in Fig. 2" as remote island sites and "Happo and Yusuhara in Fig. 2" as rural inland sites. Please note that this part was moved to the 2nd paragraph of Supplement 3, according to RC2 [5].

[16] 20,12: What are the key problems in the underestimation of NOx here? Problems with the emission dataset or chemical origin? Please further discuss that aspect with regard for using that model system in 'operational mode'. What is the ratio between NO/NO2 in total NOx?
[16] Probably lower correlation, you meant, rather than underestimation because *Sim:Obs* for $NO_x$ were not very bad, 0.92-0.94. It may be due to the emission datasets, and I personally assumed it is due mainly to the crude resolution which does not resolve the heterogeneity of emission sources. It is basically difficult to simulate primary short-lived (e.g., less than a day) species such as $NO_x$ by low resolution models. The model calculates horizontal mean concentrations but in reality the concentrations of such species should vary in space. Secondary species such as sulfate and $O_3$, or primary long-lived species (longer than a day) such as $SO_2$ are relatively easier for the crude resolution models. By the way, $NO/NO_x$ ratio in emission was assumed 0.9 in the simulation. We do not know if it is the perfect ratio, but this resulted in good model performance in terms of $NO_x$ and $O_3$ concentrations in Japan.

The relevant sentence in the 3rd paragraph of Sect. 6.2 was modified accordingly as follows:
"Low correlations of $NO_x$ were obtained probably because it is difficult to simulate primary short-lived species for the crude resolution models. The unresolvable heterogeneity of emission sources near the sites degraded the model performance of the primary species results more than they did for the secondary species such as $O_3$."

Please note that this part was moved to 2nd paragraph of Supplement 3, according to RC2.

[17] 24,20: discuss the large spread 20-100%

[17] The causes of the large spread were already discussed in the original manuscript, but it was not clear. The 5[th] paragraph of Sect. 6.2 was extensively reorganized in the revised manuscript, which is simply summarized as follows:

- Overestimation of $PM_{2.5}$ and $PM_{2.5}$(piled-up) of Bulk against those of 5-category and 3-category methods was due to the neglection of nitrate in dust by Bulk.
- Overestimation of $PM_{2.5}$ of 3-category against that of 5-cateogy was due to the unrealistic assumption of completely internal mixture of sea-salt and dust particles.

The both caused the large spread of 20-100%. Still, however, the statistical scores for the simulation-observation comparison were not different among the methods. So, we concluded that the bulk method is feasible for the operational forecast of $PM_{2.5}$, because the computational efficiency attained by the bulk method did not significantly deteriorate the model performance.

[18] 26,16: Why is that aspect particularly pronounced over sea areas? Figure 6: Why is 'Bulk' so much higher over the sea?

[18] Thank you for the good question. The contrast is prominent between land and ocean but there is also a contrast over the ocean in the top-left panel of Fig. 6 (bulk/5-ctg in spring), near from the continent (Yellow Sea and Sea of Japan) and far from the continent (East China Sea and Northwest Pacific). So, we had thought that the following mechanism is primary important: Slower deposition (of Bulk over the continent and ocean near the coast) caused longer lifetime, which in turn caused larger concentration and deposition (of Bulk) over the far downwind regions. However, it may not be true because T-$NO_3^-$ concentration of Bulk is smaller than that of Bulk in Yellow Sea and Sea of Japan. If the difference is due to the lifetime effect, surface concentration of Bulk should be larger. We came to another but very simple conclusion that the difference is due to instantaneous equilibrium assumed in Bulk. The most prominent composition of T-$NO_3^-$ over the red shaded areas in the left panels of Fig. 6 was seasalt-$NO_3$. In the presence of abundant sea-salt, $HNO_3$ quickly reacts with sea-salt in Bulk, while $HNO_3$ reacts gradually with sea-salt in 5-ctg. Consequently, $NO_3$ fractions of sea-salt mass of Bulk are larger than those of 5-ctg, as shown in the panel below (this is in spring). This caused larger $NO_3$ deposition (mostly composed of sea-salt $NO_3$) predicted by Bulk in the regions.

[Figure]

The relevant sentences (in the 2nd paragraph of Sect. 6.3 in the revised manuscript) were reorganized accordingly as follows:

"This difference is mainly due to difference in the gas-aerosol partitioning of T-$NO_3^-$, because the differences in nss-$SO_4^{2-}$ and $NH_4^+$ simulated using different methods were less significant. The dry deposition of T-$NO_3^-$ of the bulk method was smaller over land because no coarse mode $NO_3^-$ was formed over land, where its dry deposition velocity is much larger than that of submicron $NO_3^-$. Consequently, compared to the other two methods, the total (i.e., nss-$SO_4^{2-}$, T-$NH_4^+$, plus T-$NO_3^-$) dry deposition flux of the bulk method over land was smaller. On the other hand, the total dry deposition of the bulk method is larger over the ocean, where $NO_3^-$ mixed with sea salt was the major component of T-$NO_3^-$. Because the bulk method assumes instantaneous equilibrium, $HNO_3$ reacts immediately with sea salt particles, whereas $HNO_3$ gradually reacts with sea salt in the 5-category method. Consequently, $NO_3^-$ fractions of sea salt mass (as well as T-$NO_3^-$ concentrations) predicted by the bulk method are larger than those for the 5-category method, which caused larger deposition amounts of T-$NO_3^-$ (as well as the total dry deposition) predicted by the bulk method over the ocean areas."

[19] 27,4: reason for patchiness?
[19] (3rd paragraph of Sect. 6.3) Wet deposition is patcher than dry deposition because

horizontal distribution of precipitation is patchy, as we added in the revised manuscript.

*Conclusion:*

[20] As mentioned earlier, the conclusion is still missing a clear synthesis, which in places also results from missing details at various places in the manuscript. The authors are encouraged to address the following points:

P 41, Line 21: How was the operational forecast quality assessed? It is unclear if the term 'operational forecast' simply relates to the selected variables or also includes a change in the model setup (how is DA addressed?)

[20] The bulk method is faster than the other methods. If the results of fast bulk method are not far from those of the 5-category method, as a benchmark of NHM-Chem, it is successful. It is also indicated that the bulk method is eligible for operational forecasting for $PM_{2.5}$, because the computational efficiency did not significantly deteriorate the model performances in terms of the predictions of $PM_{2.5}$ surface concentrations. Of course, observation data are still not close from the results predicted by all aerosol methods. The gap between the observation and all simulations can be filled by the data assimilation or the guidance (post-process of statistical bias correction) or by the further model development. The sentences were modified accordingly. Please see the 3rd paragraph of Sect. 7 "Conclusion and discussion", in the revised manuscript.

Regarding "DA", in the previous manuscript, we had the sentence "The initial and boundary conditions should be improved before model formulation". The improvement of initial condition implicitly indicated "DA". We deleted this sentence anyway in the same paragraph, because we found it was awkward.

[21] P 41, Line 24: How exactly should the initial and boundary conditions be improved?
[21] Sorry for the confusion. We deleted the sentence which was awkward, as we replied in the previous comment [20].

[22] P 41, Line 25: Referring to your model results: where are the biggest shortcomings?
[22] We deleted the sentence, but thank you for your good question. We dare to say everything. However, the model evaluation is the only scope of the current paper, so the shortcoming is performance of the forward model in this manuscript. DA (to improve initial and boundary conditions) is out of the scope, but there are enough many DA

techniques to overcome the initial and boundary condition issues, and in addition there are statistical bias correction techniques including machine learning.

[23] P42, Line 24: See point above. Summarize dominant reasons for discrepancies.
[23] If your question is regarding the above point, "initial condition, boundary condition, or model itself?", the answer is model itself, as previously replied (in [21] and [22], and the manuscript was modified accordingly). In the model, we still don't know which are the dominant reasons. In order to get the dominant reasons, we will perform further evaluations. Accordingly, we modified the relevant sentences as follows:
"There are still large discrepancies between the simulation and observation results, but the reasons are still unclear. In order to identify the reasons and improve NHM-Chem, further evaluations should be made in the near future with respect to …"

[24] P42, Line 31: what is meant by timely and properly reflected? What are the future plans? Despite the shortcomings; what are the key benefit of the current configuration presented in this paper? What should be the core areas of future development?
[24] Timely meant immediately. Properly meant by only selecting good methods. We have two core R&D strategies, aerosol feedback and new aerosol schemes. We plan to do both, but in fact, aerosol feedback has been implemented (we haven't written a paper on it, yet). So, the future plan is to implement advanced schemes in new particle formation, secondary organics chemistry, and ice nucleation parameterizations based on recent knowledges and recent techniques. Despite the shortcomings, the key benefit of the current configuration is the comparisons of two method, fast bulk equilibrium method and accurate 5-category method, against the 3-category method, which is widely used in the air quality modeling community.

To be more specific and concrete, the last paragraph of Conclusion was separated into three and the latter two were the future plans.

---

## Author Comment (AC2) · 22 Feb 2021

Thank you very much for your constructive comments and your time for RC2. We have already revised our manuscript according to your comments and so here we attached the file containing the point-by-point responses to your comments, which will also be used for your second review. Thank you very much for your understanding.

Please also note the supplement to this comment: https://gmd.copernicus.org/preprints/gmd-2020-229/gmd-2020-229-AC2-supplement.pdf

[Figure]

**Supplement:**

Dear anonymous referee #2,

We very much appreciate your constructive comments and your time for RC2. Thanks to your review, our manuscript was substantially improved, especially for the organization of manuscript. We considered all your comments into the revised manuscript.

Point-by-point responses to your comments are written in blue in this letter.

With best regards,
Mizuo Kajino
* * *
[1] The paper of Kajino et al. compares the three aerosol representations (bulk, 3-category, 5-category) which are available in the regional CTM NHM-Chem. The results of air quality related variables (O3, NO2 etc) are compared with observations. Further, CCNs and AOT etc. are compared between the three aerosol representations. The paper is a resubmition of a paper which was rejected earlier in GMD.

I already reviewed the rejected earlier submission of the paper. The focus has now changed from a model description paper to a model evaluation paper. The manuscript fits well into the scope of the journal. The paper is very long and parts of it could be shortened, while some parts could be improved with adding some more discussion.

[1] Thank you for your evaluation and reviewing our manuscript many times. Yes, it was long. We shorten the paper by moving general description of results parts to Supplements according to your comment. Also, additional discussion is made in Sect. 7 based on your comments and in the beginning of Sect. 6 according to RC1. Please refer to RC2 [5] and RC1 [11].

[2] Further, to my opinion the language should be improved at some points. In general, the paper needs some larger revisions before it can be accepted for publication.

[2] The manuscript was sent to language editing just before submission, and so we believe that the sentences are grammatically correct. However, we admit that some parts were not clear. We tried our best to improve the language so that the sentences are clear and concrete so that readers can easily get the meaning. For example, as you commented in RC2 [9] as well as RC1 [20], "The initial and boundary conditions should

be improved before model formulation" was really awkward. We tried to remove such confusing and unclear statements in the revised manuscript.

Major comments:

[3] To my opinion the figures are partly confusing. Some of the figures show Bulk/5-cat, 3-cat/5-cat, 5-cat, other show 3-cat, 5-cat, 3-ca/5-cat. An example are Fig. 3 and Fig.4. To add a little bit of confusion the caption of Fig 4 says (same as Fig. 3). I think it would be much easier if all figures would have the same design. Similar for example Fig 8 and 6/7 or Fig 9 and Fig 10.

[3] Thank you for your comments. We agree that all figures should be harmonized. In the previous manuscript, we had Fig. 8 (fog deposition) because fog deposition scheme is a feature of NHM-Chem. However, because the manuscript was already lengthy and because the fog deposition was not pronounced in this crude grid size, we moved Fig. 8 to Supplement 5. So, previous Figs. 9-12 were shifted to Figs. 8-11. Also, it is meaningless to compare Bulk for climate-relevant variables, because Bulk was not designed to simulate them. AOT is regarded as a climate-relevant variable but, satellite AOT is often used in the data assimilation for the operational forecast of aerosol mass concentrations. Thus, the result of Bulk was also presented in Fig. 8 (AOT).

After all, all figures are harmonized but in two different ways as follows:
Figures 3-8 ($O_3$-AOT): Bulk/5-ctg, 3-ctg/5-ctg, 5-ctg
Figures 9-11 (AAOT, BC, CCN): 3-ctg, 5-ctg, 3-ctg/5-ctg

[4] Currently, there is no coupling of meteorological variables and aerosol, which is a major shortcoming of the study. This should be clearly mentioned and discussed (see also Reviewer 1).

[4] Yes, we have mentioned it in several locations but it was not appealing. Please refer RC1 [2]. We inserted the following sentences in Abstract and first paragraph of Conclusion:
(implementation of aerosol feedback processes to NHM-Chem is still ongoing)

[5] The authors included a lot of information to the supplement, but sometimes discuss this supplementary material very detailed. As an example see page 20 l5ff. Either this are supplementary information (what is fine) or this are no supplementary information. If this are no supplementary information the figures should be included into the

manuscript. If this are supplementary information the long description should be moved to the supplement to shorten the paper a little bit.

Generally, the paper offers a lot of analyses and information, but the results are of course only valid for the NHM-CHEM model. Therefore, I ask the authors to shorten parts of the general description of the results a little bit and put more empathize on the following points:

1) why are these variables important (see reviewer 1) 2) Discuss similarities /differences of the aerosol representations available in NHM-Chem with aerosol representations available in other models 3) Discuss which findings most important for other aerosol models

[5] Thank you for your useful comments. The general descriptions of the results such as comparison with observations were also presented in our previous paper (Kajino et al., JMSJ, 2019), and thus removed from the revised manuscript. Please refer to the additional statements at the end of Sect. 5 "Observations and model validations", as follows:

"Because the main objective of the paper is the aerosol module intercomparison, the general description of the results, which were already made in our previous paper (Kajino et al., 2019a), are not presented in detail here, but are presented in Supplement 3."

Accordingly, parts of 3$^{rd}$, 4$^{th}$, and 5$^{th}$ paragraph of Sect. 6.2, the entire part of 2$^{nd}$ paragraph of Sect. 6.3, and a part of last paragraph of Sect. 6.3, were moved to Supplement 3. One paragraph of Sect. 6.4 describing the comparison for the aerosol optical properties was also entirely moved to Supplement 3. A part of the first paragraph of Sect. 6.5, as well.

General features are presented in Supplement referring the main texts, but main texts do not refer materials in Supplement in the revised manuscript.

In terms of the important points as you raised, 1), 2), and 3), the point-by-point answers to the items are summarized as follows

1) WHY ARE THESE VARIABLES IMPORTANT: "Air quality variables" surface mass concentrations of $O_3$, mineral dust, and $PM_{2.5}$ are important which negatively impact the health of the population and the environment. Depositions of $SO_4^{2-}$, $NH_4^+$, and $NO_3^-$ caused environmental acidification. "Climate relevant variables" AOT, AAOT (dust and BC), CCN, and INP (CCN containing dust and BC, acting as immersion freezing) involves in aerosol-cloud-radiation interaction processes which alters energy budgets and precipitation, but are still uncertain.
→ We summarized this point in the 1st paragraph of Sect. 7

2) DISCUSS SIMILARITY/DIFFERENCES WITH OTHER MODELS: "3-category method is widely used method in air quality model such as CMAQ and WRF-Chem, and so our 5-category approach has an advantage. These aspects (external mixture of BC and dust) are already considered in many of climate models, but chemical mechanisms are usually simplified in the climate models.
→ We summarized this point in the 2nd paragraph of Sect. 7

3) DISCUSS WHICH FINDING IS MOST IMPORTANT FOR OTHER MODELS: Our message is mostly for air quality modeling communities. "Bulk method is no more used in air quality models, but still useful for operational forecast. The 3-category is less accurate when the sea-salt and dust coexist in an air mass (Gobi dust transport in Northwest Pacific or Saharan dust transport in Mediterranean or Atlantic) or when considering the light absorption and INP activity of BC and dust in the aerosol-cloud-radiation feedback system.
→ We summarized this point in the 3rd and 5th paragraphs of Sect. 7

Minor comments:

[6] P4l20ff: The discussion about what schemes/couplings have been used in which publications is were long and could be removed without loosing important information for the paper.
[6] We decided the maximum number of citations to two for each sentence. Please see the last paragraph of Sect. 2.

[7] The description in p12l5ff about the temporal length of the simulations is very confusing and should be rephrased
[7] (1st paragraph of Sect. 4.1) We deleted the dates because it is clear that "simulation

starts from January with spin-up of 5 days" means "simulation starts from 27 December". This modification enhanced the clarity of the way of time integration. Thank you for your suggestion.

[8] The sentence 'We applied the monthly mean values of GFED3 without temporal variations" is unclear (the same monthly means in each month?)
[8] (1st paragraph of Sect. 4.2) We modified it to "without daily and diurnal variations".

[9] The part in the conclusion: 'The initial and boundary conditions should be improved before model formulation. The difference in PM2.5 was large, i.e., up to 20-100%. Improving the model formulation, as well as its initial 25 and boundary conditions, is needed.' is unclear. What exactly and why should initial and boundary information be improved?
[9] Yes, the sentence was very confusing. We intended to say that data assimilation (improving initial and boundary conditions) can be used to enhance accuracy, even though the forward model is not improved. We thoroughly reorganized this paragraph of Sect. 7 "Conclusion and discussion" (3rd paragraph in the revised manuscript) as follows:
"The bulk equilibrium method was evaluated for the eligibility of operational forecast, namely, the surface mass concentrations of air pollutants such as $O_3$, mineral dust, and $PM_{2.5}$. The differences in the simulated seasonal mean concentrations between the bulk method and the 5-category method were smaller than 5% and 5-10% for $O_3$ and mineral dust, respectively. The difference in $PM_{2.5}$ was large, i.e., up to 20-100%, due to the neglection of nitrate mixed with dust particles of the bulk equilibrium method. Still, however, the statistical scores of the bulk method regarding $PM_{2.5}$ were not very different from the other methods. In order to fill the gap between observations and simulations, operational forecast is associated with data assimilation and post-process of statistical bias correction. The bulk method is not used in recent air quality models any more. Still, however, as the model performances were similar with each other, the faster bulk equilibrium method can be recommended for the use of operational forecast."
I hope this improvement is clear.

[10] I suggest to rename the conclusion into 'conclusion & Discussion'
[10] We changed it.